DOI: 10.1038/s41467-017-00892-y　　**OPEN**

# Tristetraprolin inhibits macrophage IL-27-induced activation of antitumour cytotoxic T cell responses

Qinghong Wang[1], Huan Ning[1], Hui Peng[1], Lin Wei[2], Rong Hou[1], Daniel F. Hoft[1] & Jianguo Liu[1]

IFN-γ-producing cytotoxic T lymphocytes are essential for host defense against viral infection and cancer. Here we show that the RNA-binding tristetraprolin, encoded by *Zfp36*, is needed for CD8[+] T-cell production of IFN-γ in vivo. When activated in vitro, however, IFN-γ production by naive wild type and tristetraprolin-deficient CD8[+] T-cells is comparable. IL-27 is overproduced by tristetraprolin-deficient macrophages and increased systemically in tristetraprolin-deficient mice. Tristetraprolin suppresses IL-27 production by promoting p28 mRNA degradation. Importantly, deletion of IL-27 receptor WSX-1 in tristetraprolin-deficient mice (WSX-1/tristetraprolin double knockout) leads to a reduction in cytotoxic T lymphocyte numbers. Moreover, tumor growth is accelerated, not only in tristetraprolin-deficient mice after cytotoxic T lymphocyte depletion, but also in WSX-1/tristetraprolin double knockout mice, with substantial reduction in the number of tumor cytotoxic T lymphocytes. This study describes a regulatory pathway for IL-27 expression and cytotoxic T lymphocyte function mediated by tristetraprolin, contributing to regulation of antitumour immunity.

---

[1] Division of Infectious Diseases, Allergy and Immunology, Department of Internal Medicine, Saint Louis University School of Medicine, Saint Louis University, 1100S. Grand Boulevard, St. Louis, MO 63104, USA. [2] Department of Immunology, School of Basic Medicine, Hebei Medical University, 361 East Zhongshan Road, Shijiazhuang, Hebei 050017, China. Correspondence and requests for materials should be addressed to J.L. (email: jliu9@slu.edu)

CD8[+] cytotoxic T lymphocytes (CTLs) mediate direct killing of infected, damaged, and dysfunctional cells, and are essential for clearance of viruses and for eradication of tumor cells[1]. The development of CTLs has been studied extensively, with most studies focused on how CD8[+] T-cells are activated[2]. Several cytokines, such as IL-2, IL-12, IL-21, and IL-27, induce CTLs. However, less is known about how CTLs are negatively regulated, especially at the posttranscriptional level.

Tristetraprolin (TTP) is an RNA-binding protein encoded by *Zfp36* gene, and a CCCH tandem zinc finger protein member that is involved in the post transcriptional regulation of inflammatory responses[3]. TTP binds to AU-rich elements (AREs) within the 3′ untranslated region (3′UTR), destabilizing the mRNAs encoding TNF[4], granulocyte-macrophage colony-stimulating factor[5], cyclooxygenase 2[6], interleukin-2 (IL-2)[7], IL-10[8] and the chemokine CXCL1[9], among others[10]. Overproduction of TNF, IL-17 and IL-23 in *Zfp36*[−/−] mice (also called TTP knockout (KO) mice) accounts for diseases, such as arthritis, autoimmunity and myeloid hyperplasia[11, 12]. We previously showed that TTP inhibits IL-23 expression by promoting p19 mRNA degradation via AREs in the 3′UTR[13]. Goriely and colleagues[14] later confirmed our findings and showed that IL-23 overproduction in the conventional *Zfp36*[−/−] mice causes an increase in the number of IL-17-producing T helper (Th17) cells, and that both IL-23 and IL-17A contribute to chronic inflammation in *Zfp36*[−/−] mice.

Although TTP is one of the best characterized posttranscriptional regulators and ARE binding proteins, whether TTP affects CD8[+] T-cell development and function is unclear.

IL-27 is a pleiotropic cytokine with diverse immune regulatory physiological and pathological functions, including enhancing functions of Th1 cells and CD8[+] T-cells and suppressing effects of Th2, regulatory T (Treg) cells, Th9 cells and Th17 cells[15, 16]. IL-27 transgenic mice have a high degree of CD8[+] T-cell activation[17]. In a non-cancer model, innate immune receptor agonist-based vaccine adjuvants elicit strong CD8[+] T-cell responses that are dependent on IL-27[18], illustrating the effect of endogenous IL-27. IL-27 also promotes proliferation and survival of B-cells and inhibits dendritic cell (DC) functions[15, 19]. Given its broad effects on immune regulation, IL-27 is considered a promising therapeutic target in autoimmune and infectious diseases as well as cancers. Like other members of the IL-12 family, IL-27 has two subunits, namely p28 and EBI3. IL-27 is induced by a number of TLR agonists in DCs, monocytes and macrophages, including LPS, poly (I:C), CpG and Gram negative and positive bacteria[13, 20, 21]. The p28 subunit is transcriptionally regulated by NF-κB through MyD88 and IRF1/8[22, 23]. In addition, AP-1/c-Fos/c-JNK, MAPKs and PI3K are involved in inducing p28 gene expression[24]. Furthermore, TRIF, IRF3 and IRF7 have been reported to induce IL-27 through the TLR3 and TLR4 pathways. IFN-α/β can enhance p28 expression by activating IRF1 through a

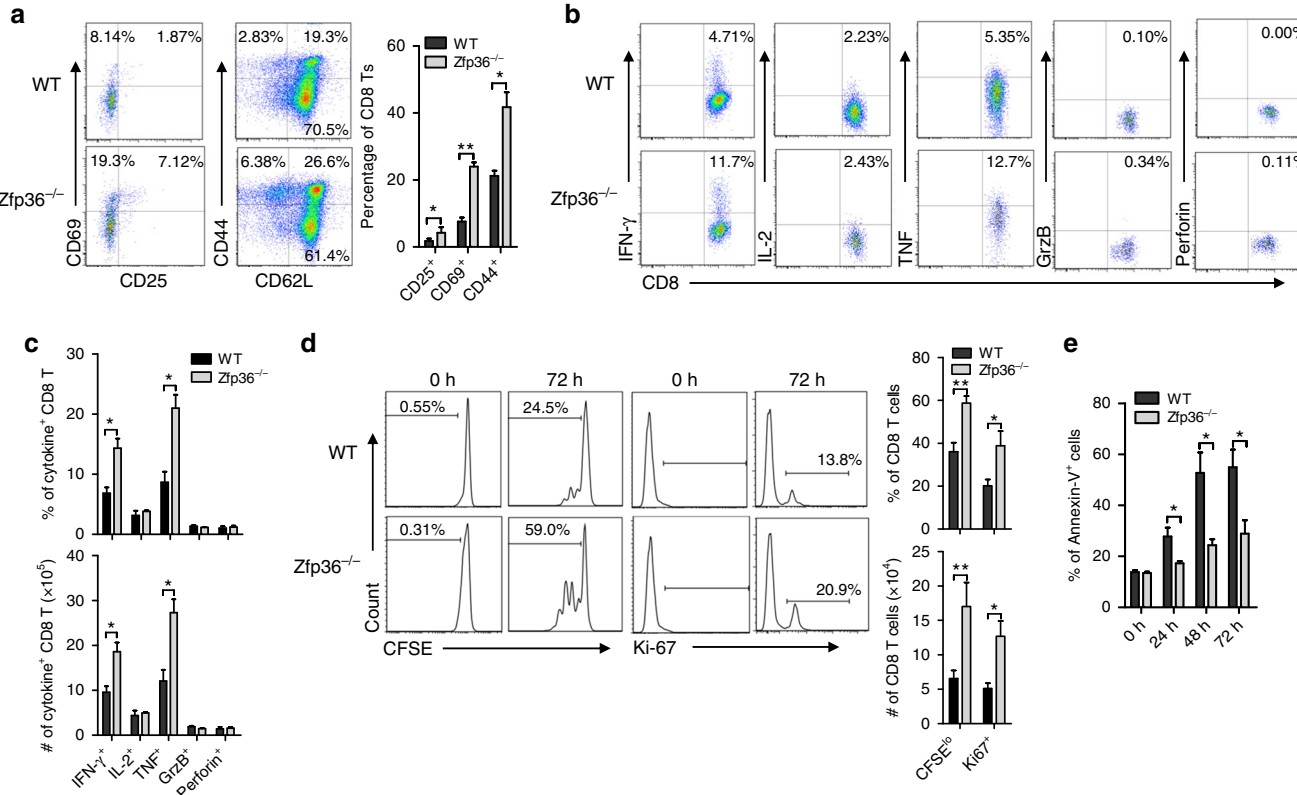

**Fig. 1** TTP deficiency promotes CD8+ T-cell activation. **a** Single splenocytes from female *Zfp36*[−/−] mice and WT littermates at 6 weeks old were stained with antibodies against CD3, CD8, CD44, CD25, CD69 and CD62L. Surface expression levels were detected with flow cytometry by gating on CD3CD8 double positive cells. Quantitative data represent means ± s.d. from three independent experiments. **b** The above splenocytes were stimulated by PMA and Ionomycin in the presence of Golgistop for 4 h, then intracellular IFN-γ, IL-2, TNF, Granzyme B and Perforin positive cells were detected with their respective antibodies, gated on CD3CD8 double positive cells. **c** Quantitative data represent means ± s.d. of cytokine[+]CD8[+] T-cells from four pair of mice as in **b**. **d** Naive CD8 T-cells isolated from spleens of WT and *Zfp36*[−/−] mice were labeled by CFSE and stimulated by plate-coated α-CD3/CD28 Abs (1 μg/ml) for 72 h. Then CFSE and Ki67 levels were detected by FACS. Quantitative data represent means ± s.d. from three independent experiments. **e** Naive CD8 T-cells isolated from spleens of WT and *Zfp36*[−/−] mice were stimulated by plate-coated α-CD3/CD28 Abs (1 μg/ml) for different times. The percentages of Annexin-V[+]apoptotic CD8 T-cells were detected by FACS and summarized as means ± s.d. from three independent experiments. Unpaired student's *t* test was used for comparison as in (**a–e**), with **$p < 0.01$; *$p < 0.05$ vs. WT control

STAT1/STAT2/IRF9 complex[25–29]. So far, most studies have focused on induction of IL-27. Little is known about the regulatory mechanisms controlling IL-27 expression.

In this study, we show that CTL numbers are increased in $Zfp36^{-/-}$ mice compared with WT mice. The induction of CTLs in $Zfp36^{-/-}$ mice is not due to increased CD8[+] T-cell differentiation, rather to enhanced production of the CD8[+] T-cell regulatory cytokine IL-27 by macrophages. TTP inhibits IL-27 expression through binding to multiple ARE sites in the p28 3′ UTR, resulting in p28 mRNA degradation. In the newly generated IL-27 receptor WSX-1 and TTP double KO mice (WSX-1/TTP double KO or DKO, thereafter), the percentages and numbers of CTLs are reduced compared with $Zfp36^{-/-}$ mice, and the suppressed tumor growth in $Zfp36^{-/-}$ mice is almost completely recovered in WSX-1/TTP double KO mice. Our data show that TTP suppresses CD8[+] T-cell function by inhibiting IL-27 production, resulting in acceleration of tumor growth. TTP may therefore be a target for cancer treatment.

## Results

**IFN-γ-producing CD8[+] T-cells are increased in $Zfp36^{-/-}$ mice.** TTP is known to inhibit the expression of cytokines and chemokines in macrophages[10]. To evaluate the effects of TTP on T-cells, we first analyzed T-cell subtypes in thymus and spleen, and found that the percentages and numbers of CD4[+] and CD8[+] T-cells in thymus (Supplementary Fig. 1a), and the percentage of

CD8[+] T-cells in spleen (Supplementary Fig. 1b) were similar between WT and $Zfp36^{-/-}$ mice, which is consistent with a previous report by Blackshear and colleagues[12]. Interestingly, we found that $Zfp36^{-/-}$ CD8 T-cells expressed higher levels of activation marker CD69, CD25 and CD44 than WT CD8 T-cells (Fig. 1a and Supplementary Fig. 1c). When comparing median fluorescence intensity (MFI) of these activation markers between naive (CD62L[+]CD44[−]) and memory (CD62L[+]CD44[+]) CD8 T-cells, however, the MFI levels of CD44, CD25, and CD69 were similar between WT and $Zfp36^{-/-}$ CD8 T-cells (Supplementary Fig. 1d), indicating an increase in effector CD8 T-cells in $Zfp36^{-/-}$ mice. Indeed, $Zfp36^{-/-}$ CD8 T-cells produced more IFN-γ and TNF, but similar levels of IL-2, granzyme B and perforin compared with WT CD8 T-cells (Fig. 1b, c). Since the effector CD8 T-cells in the mixed population were increased in $Zfp36^{-/-}$ mice (Fig. 1b, c), for a fair comparison, we next analyzed cytokine production only in CD44[+] activated memory CD8 T-cells. The percentages of cytokine-producing cells in CD44[+] activated memory cells were comparable between WT and $Zfp36^{-/-}$ CD8 T-cells (upper panel in Supplementary Fig. 1e). The numbers of CD44[+] activated memory cells producing IFN-γ and TNF were higher in $Zfp36^{-/-}$ CD8 T-cells than in WT CD8 T-cells (lower panel in Supplementary Fig. 1e). These data suggest that the increased cytokine production in $Zfp36^{-/-}$ CD8 T-cells is due to an increase in effector cells, whereas the capacity for cytokine production by $Zfp36^{-/-}$ CD8 T-cells is similar to WT cells.

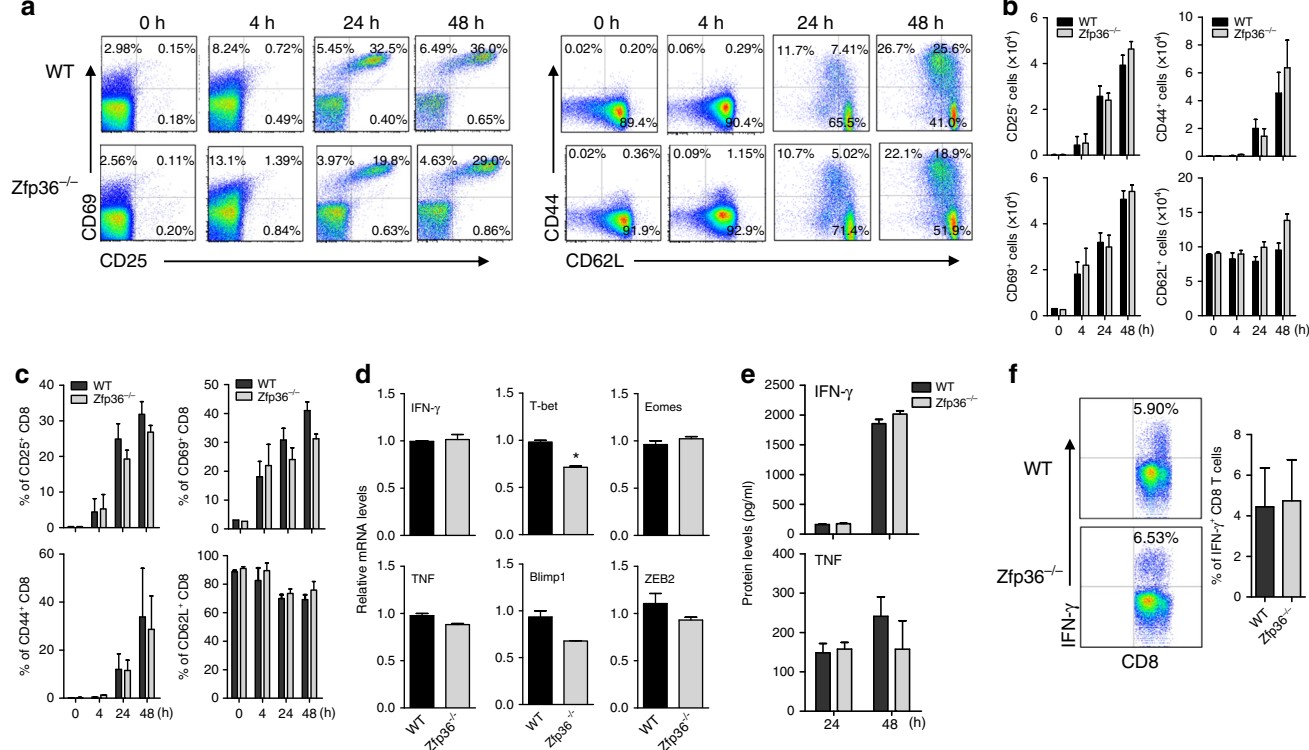

**Fig. 2** $Zfp36^{-/-}$ CD8[+] T-cells have no intrinsic defect in IFN-γ production. **a** Naive CD8 T-cells from WT and $Zfp36^{-/-}$ mice were stimulated by plate-coated α-CD3/CD28 Abs (1 μg/ml). CD69, CD25, CD44 and CD62L surface expression were detected by FACS at different time points as indicated. **b**, **c** Quantitative data represent means ± s.d. of numbers **b** and percentages **c** of CD69[+], CD25[+], CD44[+] and CD62L[+] CD8 T-cells as in **a** from three independent experiments. **d** Purified naive CD8 T-cells from WT and $Zfp36^{-/-}$ mice were stimulated by plate-coated α-CD3/CD28 Abs (1 μg/ml) for 72 h. Then the expression of IFN-γ, T-bet, Eomes, TNF, Blimp-1 and ZEB2 mRNAs were detected by qRT-PCR. The qRT-PCR data were normalized relative to GAPDH mRNA levels and further normalized to the results from WT group. Results of three independent experiments shown are means ± s.e.m. **e** IFN-γ and TNF levels in cell supernatants as in **a** were detected by ELISA. Quantitative data shown are means ± s.d. from three independent experiments. **f** Naive CD8[+] T-cells from WT and $Zfp36^{-/-}$ mice were stimulated by plate-coated α-CD3/CD28 Abs (1 μg/ml) for 3 days, rested for 3 days, and then stimulated by PMA and Ionomycin in the presence of GolGistop for 4 h. IFN-γ was detected by FACS gated on CD8 T-cells. Quantitative data shown are means ± s.d. from three independent experiments. Data are analyzed with unpaired students' t-test as in **b**–**f**, *$p < 0.05$

To evaluate the effects of TTP on CD8 T-cell proliferation and survival, we isolated naive CD8 T-cells from WT and *Zfp36*[−/−] mice and activated the cells with plate-bound anti-CD3/CD28 Abs for 3 days, then detected CFSE dilution and Ki67 expression. The percentages and numbers of CFSE[low] and Ki67[+] cells were higher in *Zfp36*[−/−] CD8 T-cells than in WT CD8 T-cells (Fig. 1d). In addition, we counted CFSE[low] cells in each division for seven generations. The difference in cell numbers reached statistical significance ($p < 0.05$, *t*-test) starting at the third generation and continued throughout the sixth generation (Supplementary Fig. 1f). Moreover, the percentages of Annexin-V[+]/PI[+] apoptotic cells were reduced in *Zfp36*[−/−] CD8 T-cells at all-time points compared with WT CD8 T-cells (Fig. 1e and Supplementary Fig. 1g). These results suggest that TTP inhibits the activation, proliferation and survival of CD8[+] T-cells.

**Zfp36[−/−] CD8[+] T-cells produce normal amounts of IFN-γ ex vivo**. To identify whether the increased effector CD8 T-cells in *Zfp36*[−/−] mice is mediated in a T-cell-intrinsic manner, we purified naive CD8[+] T-cells from *Zfp36*[−/−] and WT littermates, stimulated these cells with plate-bound anti-CD3 and CD28 antibodies in vitro, and then evaluated CD8 T-cell phenotypes at different time points. To our surprise, when naive CD8 T-cells were activated, the percentages (Fig. 2a, c) and numbers (Fig. 2b) of CD69[+], CD25[+] and CD44[+] CD8 T-cells became comparable

between WT and *Zfp36*[−/−] CD8 T-cells. The MFIs of CD44 were similar between activated WT and *Zfp36*[−/−] CD8 T-cells over a period of 48 h (Supplementary Fig. 2a). The MFIs of IFN-γ were also similar between activated/effector WT and *Zfp36*[−/−] CD8 T-cells (4702 ± 224.7 vs. 4802.5 ± 261.3; $n = 4$), when gated on the same number of CD44[+] cells. In addition, the expression of IFN-γ and TNF mRNA was comparable between activated WT and *Zfp36*[−/−] CD8 T-cells (Fig. 2d). The levels of the accumulated IFN-γ and TNF in supernatants were also comparable between WT and *Zfp36*[−/−] CD8 T-cells up to 48 h (Fig. 2e). The expression of several transcription factors important for CD8 T-cell activation, such as T-bet, Eomes, Blimp1 and ZEB2 were either no change or reduced in *Zfp36*[−/−] CD8 T-cells compared with WT CD8 T-cells (Fig. 2d). Next, we measured IFN-γ production in naive CD8[+] T-cells stimulated with anti-CD3/CD28 antibodies. As shown in Fig. 2f, there was little difference in IFN-γ production between WT and *Zfp36*[−/−] CD8 T-cells. When naive CD8[+] T-cells were stimulated with more polarizing conditions such as IL-12, IFN-γ production was increased similarly in WT and *Zfp36*[−/−] CD8 T-cells (Supplementary Fig. 2b). The expression of IFN-γ and TNF mRNA did not differ between WT and *Zfp36*[−/−] naive CD8 T-cells in response to PMA/Ionomycin and TCR stimulation at different time points (Supplementary Fig. 2c). To further confirm this observation, we used newly generated T-cell-specific TTP conditional knockout mice in which TTP[flox/flox] mice were bred with CD4-Cre mice that led to TTP being deleted

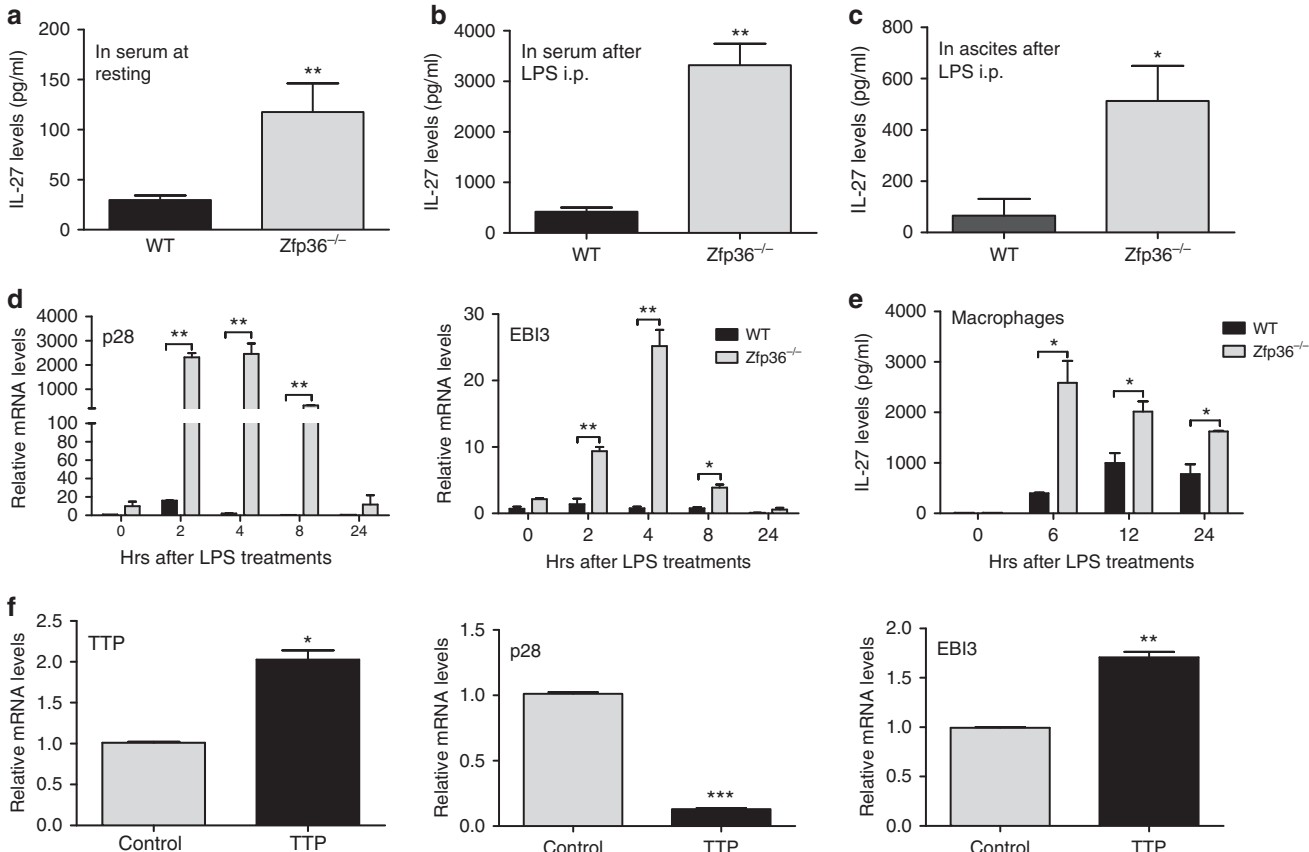

**Fig. 3** TTP inhibits IL-27 expression. **a–c** IL-27 levels in serum and ascites of WT and *Zfp36*[−/−] mice treated (7 mice per group) or not treated (3 mice per group) with LPS for 4 h were detected by ELISA. **d** Peritoneal macrophages of WT and *Zfp36*[−/−] mice were stimulated by LPS for different times as indicated. Then the mRNAs of p28 and EBI3 were detected by qRT-PCR. The data were normalized relative to GAPDH mRNA levels and further normalized to the results from WT group at 0 h ($n = 3$). **e** Culture supernatants of the cells as in **d** were used to measure IL-27 p28 protein by ELISA ($n = 3$). **f** J774A.1 cells were infected by control or TTP-expressing lentivirus for 72 h, and then stimulated by LPS for 4 h. The mRNA expression of TTP, p28 and EBI3 were detected by qRT-PCR. The data are normalized relative to GAPDH mRNA levels and further normalized to the results from control group ($n = 3$). Results shown are means ± s.d. and analyzed by unpaired student's *t*-test as in **a–f**, *$p < 0.05$; **$p < 0.01$, ***$p < 0.001$ vs. WT

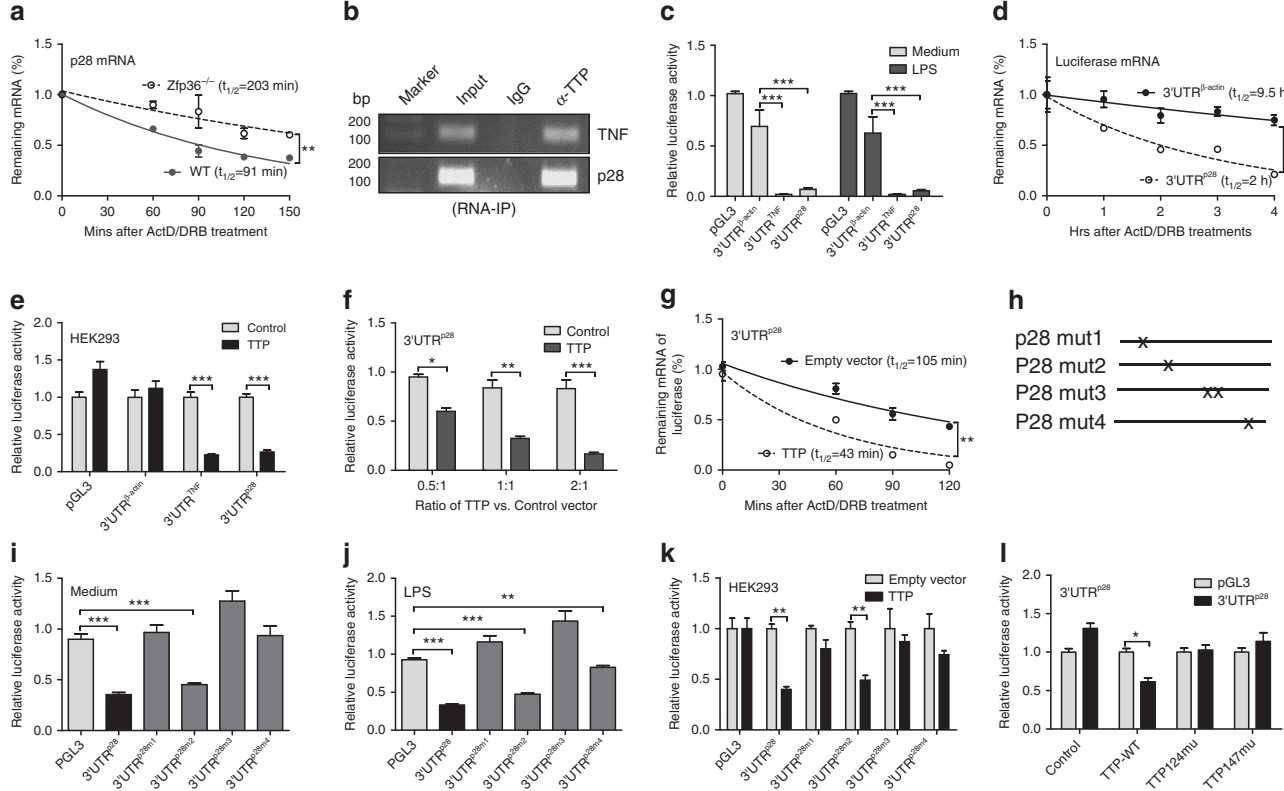

**Fig. 4** TTP regulates IL-27 p28 mRNA stability through ARE sites in the 3′UTR. **a** WT and *Zfp36*⁻/⁻ BMDMs were stimulated by LPS for 4 h. Then ActD (10 µg/ml) and DBR (5 µM) were added and RNA extracted at indicated time points. The remaining p28 mRNA levels were measured by qRT-PCR and normalized to the results at 0 min after ActD/DRB treatment (*t*-test, *n* = 3). **b** Whole cell lysates extracted from J774 cells stimulated with LPS were used for RNA-IP with anti-TTP antibody. p28 and TNF mRNA were detected by RT-PCR in RNA extracted from the IPed complex. **c** Luciferase constructs carrying the 3′UTRs of genes encoding β-actin (3′UTR^β-actin), TNF (3′UTR^TNF), p28 (3′UTR^p28) or empty vector (PGL3) was transfected into J774 cells by electroporation. Cells were treated with LPS for 7 h, and luciferase activities were measured and normalized to the activities obtained in PGL3-transfected cells without treatment (Medium) (*n* = 3). **d** Luciferase mRNA half-life was measured by qRT-PCR in cells as in **c** after ActD/DRB treatment for different times (*t*-test, *n* = 3). **e, f** p28 3′UTR luciferase construct was co-transfected with different amounts of TTP expression plasmid into HEK293 cells. Luciferase activity was detected after 48 h and normalized to the cells transfected with empty plasmid (control) (*t*-test, *n* = 3). **g** Luciferase mRNA half-life was measured in HEK293 cells transfected with TTP or control plasmids after ActD/DRB treatment for different times (*t*-test, *n* = 3). **h** Schematics of ARE mutants in the p28 3′UTR. **i, j** pGL3, p28 3′UTR, and p28 3′UTR mutant luciferase constructs were transfected into J774 cells by electroporation. Cells were treated by LPS for 24 h. Luciferase activities were measured and normalized to the activities in PGL3-transfected cells. **k** pGL3 and p28 3′UTR luciferase constructs were co-transfected with TTP or control plasmids into HEK293 cells. Luciferase activity was detected at 48 h. **l** p28 3′UTR or pGL3 luciferase constructs were co-transfected with WT and mutant TTP into HEK293 cells. Luciferase activity was measured after 48 h. One-way ANOVA with Tukey was used as in (**c, i–l**) with *n* = 3. *\*p* < 0.05; *\*\*p* < 0.01; *\*\*\*p* < 0.001 between groups

in both mature CD4 and CD8 T-cells. We isolated splenocytes from CD4^Cre^TTP^flox/flox conditional knockout mice and from CD4^Cre^TTP^+/+ control mice, and detected IFN-γ-producing CD8 T-cells after treatment with either PMA/Ionomycin or anti-CD3/CD28 antibodies for different periods. The percentages and cell numbers of IFN-γ-producing CD8 T-cells were similar between WT mice and the T-cell-specific TTP conditional knockout mice (Supplementary Fig. 2d). The ratio of naive to effector CD8 T-cells was similar between WT and CD4 conditional TTP KO mice (Supplementary Fig. 2e). These results demonstrate that CD8 T-cell activation is not altered by TTP deficiency, and cytokine production at the per cell level is similar between activated WT and *Zfp36*⁻/⁻ CD8 T-cells, suggesting that TTP regulates CD8 T-cell proliferation and survival in an intrinsic manner, while modulates cytokine production in an extrinsic fashion.

**IL-27 production is increased in *Zfp36*⁻/⁻ macrophages.** It has been reported that IL-27 through its receptor IL-27Ra activates STAT1 and STAT3 that induce T-box transcription, leading to

proliferation and activation of CD8 T-cells[30, 31]. Therefore, we evaluated whether IL-27 production was affected by TTP in an endotoxin shock murine model. The levels of IL-27 in serum were increased significantly in *Zfp36*⁻/⁻ mice compared with WT mice, both at the resting state (Fig. 3a) and after LPS challenge (Fig. 3b), as well as in ascites (Fig. 3c), liver and lung (Supplementary Fig. 3a, b). We previously reported that IL-27 is mainly produced by macrophages through TLR4 signaling in a MyD88 dependent manner[22, 23]. Then, we measured the expression of p28 and EBI3 subunits in activated peritoneal macrophages. Both p28 and EBI3 mRNA expression were significantly increased in *Zfp36*⁻/⁻ macrophages compared with WT macrophages after LPS treatment, with the increase of p28 more than 100 fold higher than the increase of EBI3 (Fig. 3d). This is consistent with our previous reports that p28 subunit is the limiting factor for producing biological IL-27[22]. IL-27 protein levels were also markedly increased in culture supernatants of *Zfp36*⁻/⁻ macrophages compared with WT macrophages (Fig. 3e). In addition, the increased IL-27 production was also observed in *Zfp36*⁻/⁻ bone marrow derived macrophages (Supplementary Fig. 3c). In line

with these results, overexpression of TTP by lentiviral (Fig. 3f) and by adenoviral (Supplementary Fig. 3d) transduction inhibited the expression of p28, but not EBI3 in macrophages stimulated by LPS. Taken together, these data demonstrate that TTP inhibits the production of the CD8 T-cell regulatory cytokine IL-27.

**TTP regulates p28 mRNA stability through ARE sites.** It has been reported that TTP induces degradation of targeting genes through AREs in the 3′UTR[10]. We checked the 3′UTR of p28 mRNA and found that there are five putative ARE sites. To determine whether TTP affects p28 mRNA stability, we measured p28 mRNA half-life after blocking de novo RNA synthesis with ActD and DRB. The half-life of p28 mRNA was increased more than two-fold in *Zfp36*[−/−] macrophages compared with WT cells (Fig. 4a), indicating a posttranscriptional regulation of p28 mRNA by TTP. Next, we performed RNA-IP with anti-TTP antibody, and found that anti-TTP antibody but not epitope control IgG pulled down the p28 mRNA (Fig. 4b), demonstrating

a direct binding of TTP to p28 mRNA. As 3′UTR plays an important role in controlling mRNA stability, we cloned the p28 3′UTR immediately downstream of luciferase gene and transfected it into J774 cells, followed by LPS stimulation. Because TTP is known to target TNF 3′UTR for degradation and β-actin mRNA is quite stable, we included TNF 3′UTR and β-actin 3′UTR as positive and negative controls, respectively. The luciferase activity was significantly reduced in cells transfected with the p28 3′UTR, at similar levels of reduction as the TNF 3′UTR positive control (Fig. 4c). To confirm that the reduced luciferase activity was due to 3′UTR-mediated mRNA decay, we measured the half-life of luciferase mRNA, and found that the half-life of luciferase mRNA was significantly reduced in cells transfected with the p28 3′UTR compared with the β-actin 3′UTR transected cells (Fig. 4d). HEK293 cells are sensitive to ectopically expressed TTP, because they express minimal amounts of endogenous TTP. Therefore, we introduced TTP and p28 3′UTR luciferase vector into these cells, followed by measuring luciferase activity. TTP dose-dependently inhibited luciferase activity in cells transfected

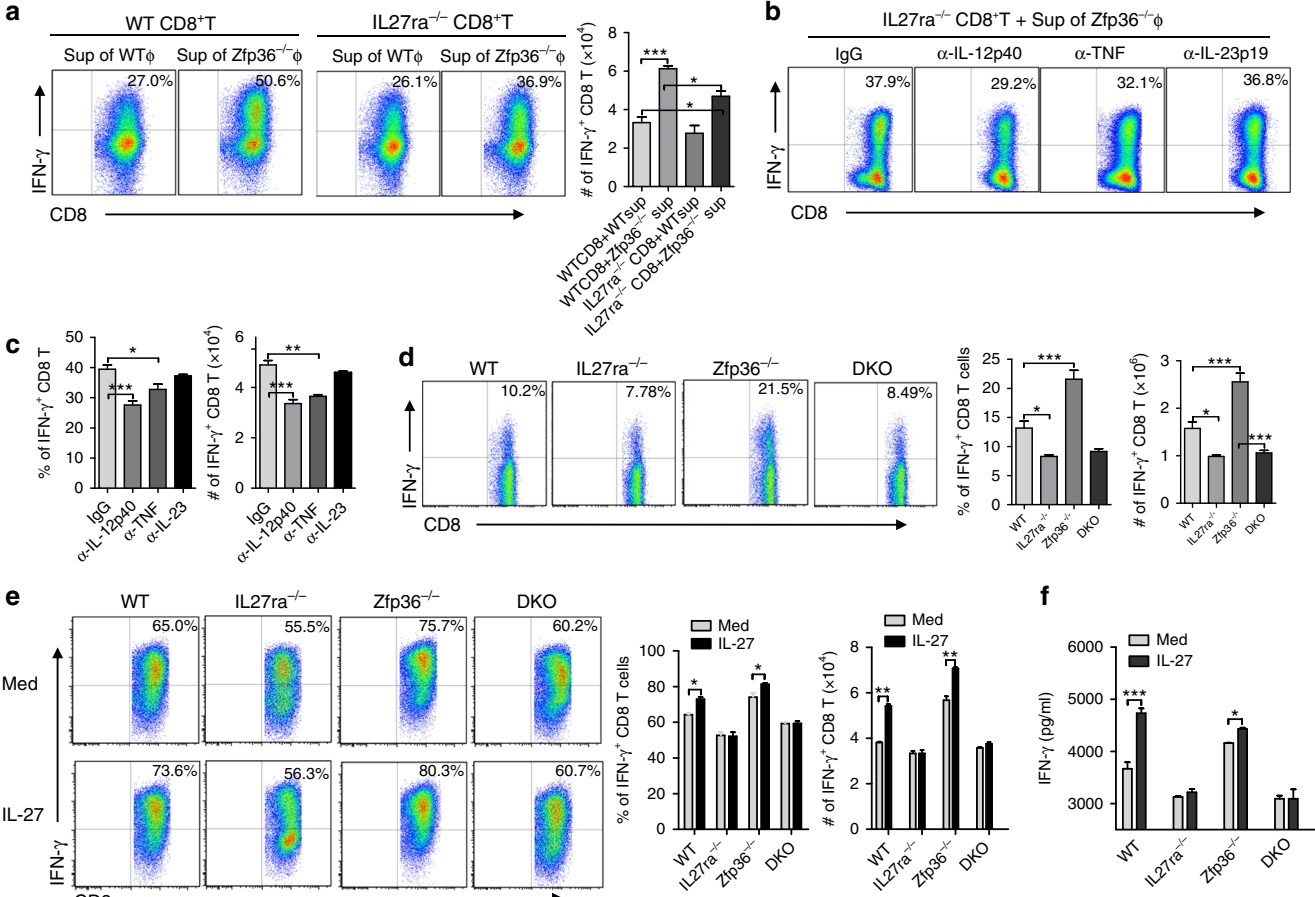

**Fig. 5** IL-27 derived from *Zfp36*[−/−] macrophages enhances IFN-γ production by CD8+ T-cells. **a** WT and *Zfp36*[−/−] peritoneal macrophages were stimulated by LPS (1 μg/ml) for 24 h, and then supernatants were collected. Purified naive WT and *IL27ra*[−/−] CD8[+] T-cells were cultured with the supernatants (1:1) in the presence of plate-coated α-CD3/CD28 Abs (1 μg/ml) for 3 days. IFN-γ[+]CD8[+]T-cells were detected by FACS. Quantitative data shown are means ± s.d. from three independent experiments. **b**, **c** Supernatants of the above *Zfp36*[−/−] peritoneal macrophages were respectively pre-treated with neutralizing antibodies against IL-12 p40, TNF and IL-23 p19 (10 μg/ml) for 30 min, then cultured with *IL27ra*[−/−] naive CD8 T-cells as in **a**. IFN-γ[+]CD8[+]T-cells were detected by FCM. Quantitative data shown are means ± s.d. from three independent experiments. **d** Splenocytes isolated from WT, *IL27ra*[−/−], *Zfp36*[−/−] and *IL27ra*[−/−]*Zfp36*[−/−] double KO (DKO) mice were stimulated by PMA and Ionomycin with GolgiStop for 4 h. IFN-γ[+] cells were detected by FACS, gated on CD3[+]CD8[+] double positive cells. Quantitative data represent means ± s.d. from three mice of each group. **e** Splenocytes of WT, *IL27ra*[−/−], *Zfp36*[−/−] and DKO mice were stimulated by soluble α-CD3/CD28 Abs (1 μg/ml) in the presence or absence of IL-27 (2 ng/ml) for 3 days. Then IFN-γ[+]CD8[+]T-cells were detected by FCM. Quantitative data represent means ± s.d. from three independent experiments. **f** IFN-γ levels in the supernatants as in **e** were detected by ELISA. Data were analyzed with one-way ANOVA (Nonparametric) with Tukey (compare all pairs of columns), *p < 0.05; **p < 0.01; ***p < 0.001. Φ: macrophage; sup: supernatant

with p28 3′UTR (Fig. 4e, f). The half-life of luciferase mRNA was also shortened by TTP in cells transfected with the p28 3′UTR construct (Fig. 4g). These results indicate that p28 3′UTR controls its mRNA stability. To determine which AREs are important for p28 mRNA decay, we generated four ARE mutants as illustrated in Fig. 4h, transfected these mutants into J774 cells, and then measured luciferase activity. The p28 3′UTR-mediated inhibition of luciferase activity was completely abolished in cells transfected with all mutant p28 3′UTR constructs except the mutant#2 (Fig. 4i, j), suggesting that TTP through multiple AREs promotes p28 mRNA decay. Similar results were also shown in HEK293 cells overexpressing TTP (Fig. 4k). TTP contains a CCCH tandem zinc-finger domain. Zinc fingers of this type have been found in many RNA-binding proteins, and are responsible for binding to the 3′UTRs of target mRNAs[4, 32]. To determine

their contribution to p28 mRNA decay, we co-transfected wild type TTP, TTP124 mutant and TTP147 mutant with the p28 3′UTR-luciferase vector, then measured luciferase activity. As shown in Fig. 4l, TTP-mediated inhibition of luciferase activity was lost in cells transfected with both mutants, indicating that these two zinc fingers are required for TTP-mediated degradation of p28 mRNA.

**IL-27 mediates the IFN-γ production by *Zfp36*$^{-/-}$ CD8+ T-cells.** To determine the role of IL-27 in IFN-γ production by *Zfp36*$^{-/-}$ CD8$^+$ T-cells, we purified naive CD8$^+$ T-cells from WT and *IL27ra*$^{-/-}$ mice, cultured these cells with culture supernatants collected from activated WT or *Zfp36*$^{-/-}$ peritoneal macrophages, and then measured IFN-γ production. The supernatants of

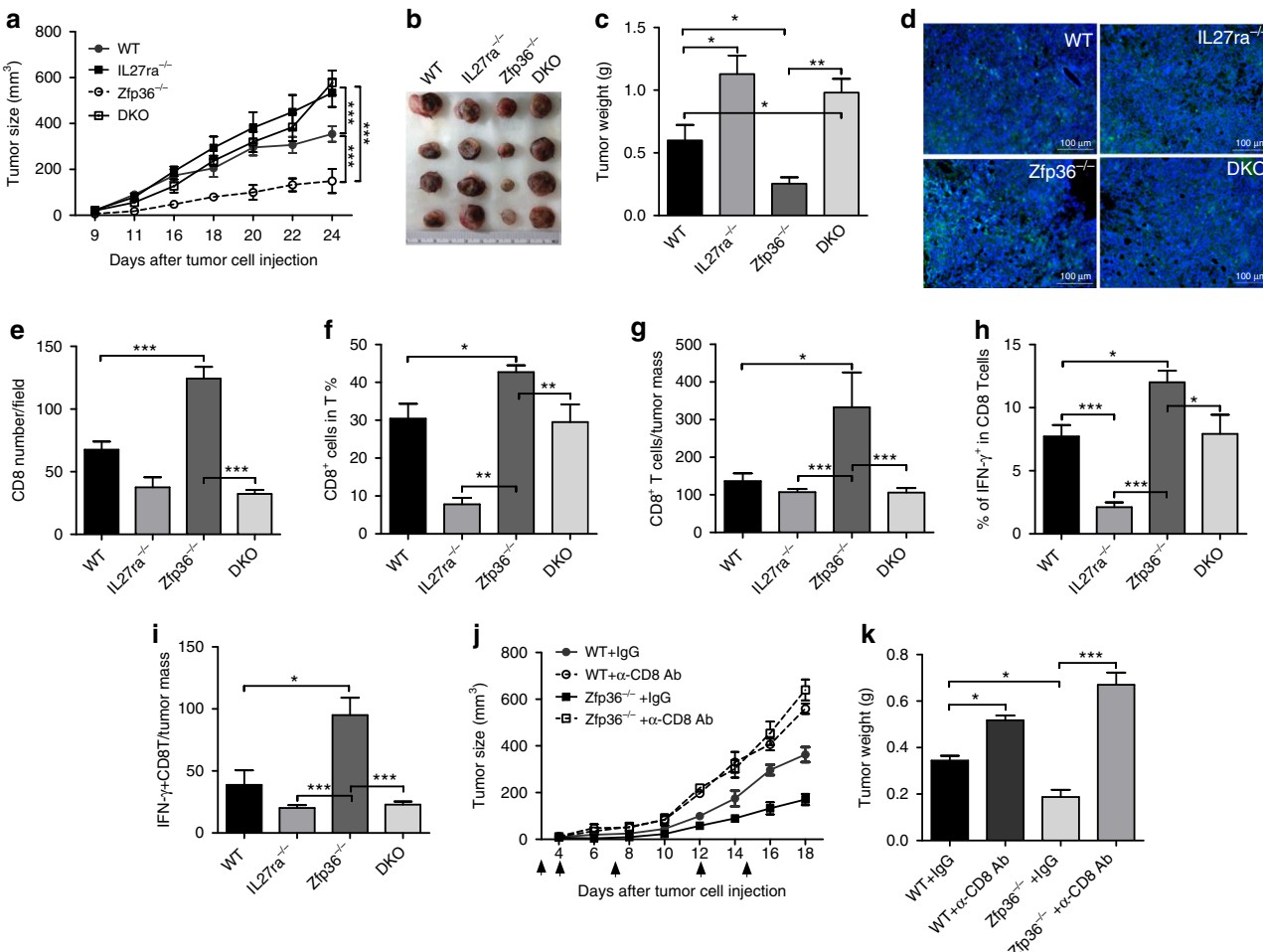

**Fig. 6** TTP drives tumor progression by inhibiting IL-27 production and CTL function. **a** EO771 cells were inoculated into mammary pads of female WT, *IL27ra*$^{-/-}$, *Zfp36*$^{-/-}$ and DKO mice aged 6–8 weeks old. Tumor size was measured in three dimensions. Each group has four mice. **b, c** Mice were killed 24 days after tumor cell inoculation; tumors were imaged, weighted, and shown as means ± s.d. (four tumors in each group). **d** Tumor tissue sections were sainted with rat anti-CD8 antibody, AF488-labeled anti-rat IgG (1:200, *green*), and diaminidophenylindol (DAPI, 1:200, blue). The images were taken by fluorescence microscopy (magnification: 100×). Rat IgG was used as negative control. **e** The numbers of CD8$^+$ cells as in **d** were counted per field on five different sections of each tumor and summarized as means ± s.d ($n = 4$). **f** The percentages of CD8$^+$ T-cells in tumor homogenates were detected by FACS, gated on CD3$^+$CD8$^+$ double positive cells ($n = 4$). **g** The numbers of CD8$^+$ T-cells per tumor weight were calculated and presented as CD8 numbers per gram of tumors ($n = 4$). **h** The percentages of IFN-γ$^+$ cells in tumor homogenates were determined by FACS, gated on CD3$^+$CD8$^+$ double positive cells ($n = 4$). **i** The numbers of IFN-γ$^+$CD8$^+$T-cells per tumor weight were calculated and presented as the numbers per gram of tumors ($n = 4$). **j** EO771 cells were inoculated into mammary pads of eight female WT and *Zfp36*$^{-/-}$ mice. Half of WT and *Zfp36*$^{-/-}$ mice were treated by giving CD8 depletion antibody (150 μg/mouse, *InVivo*MAb anti-mouse CD8α) or control IgG *i.p* at 0, 4, 7, 12 and 15 day after tumor cell inoculation for five times in total (indicated as ↑). Tumor sizes were measured in three dimensions. Each group has four mice. **k** Mice were sacrificed 18 days after tumor cell inoculation, and tumors were taken out and weighted **g**. Data shown are means ± s.d. with four tumors in each group. All data were analyzed with One-way ANOVA (Nonparametric) with Turkey (Compare all pairs of columns), *$p < 0.05$; **$p < 0.01$; ***$p < 0.001$ between indicated groups

$Zfp36^{-/-}$ macrophages induced robust IFN-γ production in WT CD8 T-cells compared with the supernatants of WT macrophages (Fig. 5a and Supplementary Fig. 4a). Importantly, when culturing with $IL27ra^{-/-}$ CD8 T-cells, less IFN-γ was produced (Fig. 5a and Supplementary Fig. 4a), indicating that IL-27 produced by $Zfp36^{-/-}$ macrophages contributes to the higher IFN-γ production in CD8 T-cells of the $Zfp36^{-/-}$ mice. It is worth noting that supernatants of $Zfp36^{-/-}$ macrophages were able to enhance IFN-γ production even without $IL27ra$ (Fig. 5a), indicating that other cytokines produced by $Zfp36^{-/-}$ macrophages may contribute to IFN-γ production. TNF and IL-23 are known to be targeted by TTP[13, 14] and able to activate CD8 T-cells[33]. To tease this out, we treated the $Zfp36^{-/-}$ supernatants with TNF and IL-23 p19 neutralizing antibody prior to culturing it with WT naive CD8 T-cells. Neutralizing TNF reduced IFN-γ production in WT CD8 T-cells induced by the $Zfp36^{-/-}$ supernatants (Supplementary Fig. 4b). However, a significant amount of IFN-γ (>40%) was still produced after TNF and IL-23 blockade (Supplementary Fig. 4b). IL-12 is known to be a strong activator for CD8 T-cells. To determine whether other cytokines (such as IL-12, TNF and IL-23) may act with or without IL-27 to activate CD8 T-cells, we co-cultured the $IL27ra^{-/-}$ CD8 T-cells with supernatants from activated $Zfp36^{-/-}$ macrophages in the presence of neutralizing antibodies against IL-12/IL-23 p40, TNF, and IL-23 p19. In the absence of IL-27 signaling, neutralizing IL-12 and TNF, but not IL-23, inhibited the production of IFN-γ in $IL27ra^{-/-}$ CD8 T-cells (Fig. 5b, c), indicating that IL-12 and TNF induce IFN-γ production in CD8 T-cells independent of IL-27 signaling. These data also demonstrate that IL-27 promotes IFN-γ production in CD8 T-cells mediated by $Zfp36^{-/-}$ supernatants, and other cytokines, such as IL-12 and TNF, may work with IL-27 to further enhance IFN-γ production. To further confirm the role of IL-27 in IFN-γ production by $Zfp36^{-/-}$ CD8 T-cells, we generated TTP and WSX-1 double knockout mice (TTP/WSX-1 DKO), isolated splenocytes from WT, $IL27ra^{-/-}$, $Zfp36^{-/-}$ and TTP/WSX-1 DKO mice, and measured IFN-γ production. IFN-γ-producing CD8 T-cells were increased in $Zfp36^{-/-}$ mice, and this increase was diminished in TTP/WSX-1 DKO mice (Fig. 5d). Next, we activated these splenocytes with recombinant IL-27 in the presence of anti-CD3 antibody, and found that IL-27 increased IFN-γ production in CD8 T-cells from both WT and $Zfp36^{-/-}$ mice, but not in CD8 T-cells from $IL27ra^{-/-}$ mice or TTP/WSX-1 DKO mice (Fig. 5e, f), further indicating that IL-27 produced by $Zfp36^{-/-}$ macrophages enhances IFN-γ production by CD8 T-cells of the $Zfp36^{-/-}$ mice.

**IL-27 and CTLs mediate antitumour effects of $Zfp36^{-/-}$ mice.** To determine the effects of TTP-mediated IL-27 inhibition on tumor development, we employed a murine mammary gland tumor model by inoculating E0771 cells into mammary gland pads of WT, $Zfp36^{-/-}$, $IL27ra^{-/-}$ and TTP/WSX-1 DKO mice, followed by monitoring of tumor growth. As shown in Fig. 6a, tumors grew slower in $Zfp36^{-/-}$ mice than WT mice. Importantly, the retarded tumor growth in $Zfp36^{-/-}$ mice was completely relieved in TTP/WSX-1 DKO mice. The reduced sizes and weights of tumors in $Zfp36^{-/-}$ mice were also recovered in the DKO mice (Fig. 6b, c). These data indicate that TTP-mediated pro-tumor effects act through inhibition of IL-27 signaling pathway. By analyses of tumor sections stained with H&E, we observed more tumor infiltrating lymphocytes in tumors of the $Zfp36^{-/-}$ mice compared with other groups (Supplementary Fig. 5a).

To better visualize infiltrates, we stained and quantified CD8 T-cells on different tumor sections in each group. More infiltrating CD8 T-cells were observed per field in tumors of

$Zfp36^{-/-}$ mice compared with the tumors of other groups; and the increased CD8 T-cells were reduced significantly in tumors of the DKO mice (Fig. 6d, e). Next, we isolated mononuclear cells from tumors and analyzed the infiltrating CD8 T-cells. There were increased percentages (Fig. 6f) and numbers of CD8+ T-cells (Fig. 6g) in the tumors of $Zfp36^{-/-}$ mice compared with other tumors. Similarly, percentages of CTLs increased in tumors of $Zfp36^{-/-}$ mice compared with other groups (Fig. 6h). In addition, absolute numbers of CTLs were significantly increased in tumors of $Zfp36^{-/-}$ mice compared with tumors of other mice, when normalized against tumor mass (Fig. 6i), indicating an important role for CD8 T-cells in antitumor activity of the $Zfp36^{-/-}$ mice. To define whether CD8 T-cells mediate the resistance to the tumor cell challenge of $Zfp36^{-/-}$ mice, we depleted CD8 T-cells with CD8 depletion antibody and monitored tumor growth. Consistent with the results in Fig. 6a, $Zfp36^{-/-}$ mice showed retarded tumor growth compared with WT control mice (Fig. 6j). More importantly, depletion of CD8 T-cells in $Zfp36^{-/-}$ mice accelerated tumor growth (Fig. 6j) and increased tumor mass (Fig. 6k). We checked CD8 T-cell depletion efficiency on day 18 after tumor cell inoculation. The percentages of CD8 T-cells in spleens were significantly reduced after administration of CD8 depletion antibody (Supplementary Fig. 5b). Taken together, these results demonstrate that IFN-γ-producing CD8 T-cells mediate the antitumor effects in $Zfp36^{-/-}$ mice dependent on IL-27.

**Discussion**

CTLs are critical in host defense against viral infection and cancer, and their differentiation depends on signals triggered by cytokines, especially IL-12 family cytokines. IL-27 is one of the IL-12 cytokine family members and known to play an important role in CTL development[34]. We previously demonstrated that IL-27 production by macrophages is mediated by TLR4 signaling through adapter protein MyD88[22]. In this study, we demonstrate for the first time that TTP suppresses IL-27 production by targeting the p28 mRNA for degradation, resulting in suppression of CD8 T-cell functions.

TTP, one the best characterized RNA-binding proteins, is important in inhibiting the expression of proinflammatory cytokines in macrophages. Its effects on CD8 T-cells, however, are unknown. We found that effector CD8 T-cells are increased in $Zfp36^{-/-}$ mice (Fig. 1a and Supplementary Fig. 1c), indicating a negative regulation of CD8 T-cell activation by TTP. Among CD8 T-cell effector molecules, IFN-γ and TNF production increased, while Granzymes and Perforin remained unchanged in $Zfp36^{-/-}$ CD8 T-cells compared with WT-cells (Figs. 1b, c), suggesting that TTP modulates CD8 T-cell functions by selectively regulating expression of certain effector molecules. The increased CFSE dilution and Ki67 of naive CD8 T-cells deficient of TTP (Fig. 1d & Supplementary Fig. 1f) suggest an inhibitory effect of TTP on CD8 T-cell proliferation. TTP is reported to regulate IL-2 expression[7] that may contribute to the proliferation of $Zfp36^{-/-}$ CD8 T-cells (Fig. 1d). However, the similar levels of IL-2 between WT and $Zfp36^{-/-}$ CD8 T-cells (Fig. 1b, c), suggests that the increased proliferation of $Zfp36^{-/-}$ CD8 T-cells is independent of IL-2 and may be mediated by other factors. In addition to its effects on cell proliferation, TTP also affects CD8 T-cell survival, as $Zfp36^{-/-}$ CD8 T-cells were less likely to die compared with WT CD8 T-cells (Fig. 1e and Supplementary Fig. 1g). TTP is known to inhibit tumor cell growth by causing cell cycle arrest[35–37]. Whether TTP regulates CD8 T-cell proliferation through similar mechanisms as the tumor cells needs to be further investigated.

TTP appears to suppress the functions of CD8 T-cells by inhibiting cell activation and cell growth. Though $Zfp36^{-/-}$ CD8

T-cells produced more IFN-γ and TNF in vivo, the effector CD8 T-cells differentiated in vitro from naive *Zfp36*[−/−] CD8 T-cells displayed similar effector phenotype and produced similar amounts of IFN-γ as the WT CD8 T-cells (Fig. 2), suggesting that *Zfp36*[−/−] CD8 T-cells have no defect in producing cytokines. Blackshear's group reported that T-cell development is normal in *Zfp36*[−/−] mice[38]. We also found comparable CD4 and CD8 T-cells in thymus, as well as similar percentages of CD8[+] T-cells in spleen between WT and *Zfp36*[−/−] mice (Supplementary Fig. 1a, b). In addition, Molle et al[14]. reported that the development of Th1, Th17 and Th22 cells in *Zfp36*[−/−] mice were normal. Collectively, these data indicate that TTP does not affect IFN-γ and TNF production in CD8 T-cells. The culture supernatants of *Zfp36*[−/−] macrophages increased IFN-γ production in WT but not *IL27ra*[−/−] CD8 T-cells further confirms that IL-27 produced by *Zfp36*[−/−] macrophages contributes to CD8 T-cell activation. There was still enhanced IFN-γ production by supernatants from *Zfp36*[−/−] macrophages even without WSX-1 (Fig. 5a), indicating cytokines other than IL-27 may also contribute to IFN-γ production. To understand the association of other cytokines (IL-12, TNF and IL-23) with IL-27 in activating CD8 T-cells, we used *IL27ra*[−/−] CD8 T-cells. Neutralization of IL-12 and TNF, but not IL-23, inhibited IFN-γ production in *IL27ra*[−/−] CD8 T-cells (Fig. 5b, c), indicating that IL-12 and TNF induce IFN-γ production in CD8 T-cells independent of IL-27 signaling. These data also demonstrate that IL-27 produced by TTP deficient macrophages promotes IFN-γ production in CD8 T-cells, and other cytokines, such as IL-12 and TNF, may work with IL-27 to further enhance IFN-γ production.

IL-27 is a pleiotropic cytokine important for CTL generation and function[34]. A recent study by Kedl's and colleagues[18] reported that subunit vaccines combined with various TLR agonists as adjuvant induce robust CD8 T-cell responses dependent on IL-27, which is in contrast to IL-27-independent infectious challenge. So far, most studies have focused on how IL-27 is induced, and little has been done on the regulatory pathways important for IL-27 inhibition. As an RNA-binding protein, TTP is known to suppress the expression of proinflammatory cytokines by binding directly to AREs in the 3′UTR of their mRNAs, leading to their deadenylation and decay. Kovarik's group recently performed a transcriptome-wide study to identify potential TTP targets by analyzing the consensus binding sequence of TTP. In their TTP binding site atlas (http://ttp-atlas.univie.at), 498 genes contain the TTP binding sites. Though IL-27 is on the list, their work did not show IL-27 mRNA is targeted by TTP. Consistent with the biological function of TTP, our data demonstrate that TTP inhibits IL-27 production by targeting p28 mRNA for degradation via direct binding to AREs in the 3′UTR of p28 mRNA. Among five putative AREs in p28 3′UTR, four appeared to be involved in p28 mRNA degradation by TTP, except the second ARE. TTP is a CCCH tandem zinc finger protein, and mutation of any one of the 8 CCCH residues results in complete loss of binding activity[10]. Indeed, two TTP zinc finger mutants completely lost suppressive activity on p28 3′UTR-mediated luciferase activation (Fig. 4i), confirming the requirement for the two zinc fingers in p28 mRNA decay. IL-27 induces IL-10-producing CD4 T-cells[39–41]. As a potent anti-inflammatory cytokine, IL-10 also inhibits IL-27 production during infection-associated inflammation[42]. This raises the question whether IL-10 impacts the regulatory pathway of TTP on IL-27. We stimulated WT macrophages with LPS, IL-10 or both, then measured IL-27 p28 and EBI3 mRNA expression. IL-10 inhibited the p28 (Supplementary Fig. 6a) but not EBI3 (Supplementary Fig. 6b) mRNA expression induced by LPS. In the same setting, LPS strongly induced TTP expression which is in line with our previous report[13]. While IL-10 induced minimal TTP expression by itself,

it inhibited LPS-induced TTP expression (Supplementary Fig. 6c). IL-10 does not upregulate TTP in activated macrophages, suggesting that IL-10-mediated suppression of IL-27 is independent of TTP. Furthermore, we treated WT and *Zfp36*[−/−] BMDMs with LPS in the presence or absence of IL-10 and then measured p28 mRNA expression. IL-10 inhibited p28 expression at similar levels in both BMDMs (Supplementary Fig. 6d), indicating that IL-10 does not impact this pathway.

Tumor infiltrating CD8[+] T-cells play a critical role in tumor eradication, including breast tumor[43–47]. IL-27 exerts antitumor effects by affecting immune cell functions and cytokine production[48], and by promoting the generation, proliferation, survival, and function of CTLs[31]. Overexpression of IL-27 in Colon26 cells induced CTL development in tumor-bearing mice[49, 50]. Local delivery of IL-27 to tumors significantly decreased metastasis and growth of neuroblastoma and CT26 tumors by inducing tumor specific CTL[51–53]. Increased tumor infiltrating CD8 T-cells have been also shown to be significantly associated with better prognosis in patients with ER negative, HER-2 negative and triple-negative breast cancer subtypes. We found that TTP negatively regulated CTL function through inhibiting IL-27 production (Figs. 1, 5). In a murine mammary gland tumor model, mice deficient of TTP showed retarded tumor growth, which is in line with the increased tumor-infiltrating CD8[+] T-cells of *Zfp36*[−/−] mice (Fig. 6d–i). Depletion of CD8 T-cells in *Zfp36*[−/−] mice resulted in accelerated tumor growth (Fig. 6j, k), demonstrating that antitumor immunity in *Zfp36*[−/−] mice depends on CD8 T-cells. The retarded tumor growth in *Zfp36*[−/−] mice was recovered in the TTP and WSX-1 double knockout mice, further suggesting that IL-27-dependent antitumor effect is mediated by CD8 T-cells in *Zfp36*[−/−] mice. Kratochvill et al.[54] found that macrophages deficient of TTP suppressed tumor growth, probably through regulating the function of tumor-associated macrophages.

In summary, this study unveils a novel regulatory pathway for IL-27 expression and CTL function mediated by TTP, which contributes to the regulation of antitumor immunity. It also indicates that TTP expressed inside host macrophages can dampen antitumor immunity through inhibition of innate and adaptive immune responses. Considering the antitumor effects of the TTP expressed inside tumor cells[32], target expression of TTP specifically in tumor cells should be considered when trying to use TTP as a tumor suppressor.

## Methods

**Mice.** *Zfp36*[+/−] and *Zfp36*[flox/flox] mice were kindly provided by Blackshear and colleagues[55] (National Institute of Environmental Health Sciences). *IL27ra*[−/−] mice were obtained from Jackson laboratory. TTP and WSX-1 double deficient mice were generated by crossing *Zfp36*[+/−] mice and *IL27ra*[−/−] mice. CD4[Cre]TTP[f/f] mice were obtained by crossing mice expressing Cre recombinase under the control of the murine CD4 promoter (CD4[cre]) mice purchased from the Jackson Laboratory (Bar Harbor, ME)[56]. All mice were on the C57BL/6 background and were bred at the animal facility of Saint Louis University. Mice used in the study were female aged 6–8 weeks old because of their relevance to breast cancer. Animal experiments were approved by the Institutional Animal Care and Use Committee at Saint Louis University and were performed according to federal and institutional guidelines under IACUC protocol number 2046.

**Cells and reagents.** RAW264.7 and J774A.1 cells, as well as HEK293 cells were obtained from the American Type Culture Collection. All cell lines were authenticated by STR DNA profiling and detected for mycoplasma contamination by PCR DNA analysis. Mouse peritoneal macrophages were obtained by lavage 3 days after injection of sterile 3% thioglycolate broth (0.5 ml i.p. per mouse) and plated in 24 well tissue culture plates ($1 \times 10^6$ cells/well) with RPMI1640 complete medium. Mouse bone marrow-derived macrophages (BMDMs) were generated from bone marrow cells and cultured with complete RPMI1640 medium containing 10 ng/ml of M-CSF for 1 week[13]. Anti-mouse TTP antibody and LPS (*Escherichia coli* 0217: B8) were purchased from Sigma-Aldrich (St. Louis, MO). Recombinant mouse IL-27 was purchased from PeproTech (Rocky Hill, NJ). Neutralizing antibodies against IL-12/23 p40 (clone: C17.8; Cat #: 505304; Biolegend), IL-23 p19 (clone:

MMp19B2; Cat #: 513805; Biolegend), and TNF (clone: TN3-19.12; Cat #: 14-7423-85; eBioscience) were purchased from Biolegend and eBioscience, respectively.

**T-cell purification and culture**. Naive CD8[+] T-cells were purified by EasySep™ Mouse Naive CD8[+] T-cell Negative Isolation Kit (Catalog#: 19858, Stemcell Technologies) from splenocytes of WT or knockout mice, and co-cultured with supernatants collected from activated macrophages in the presence of plate-bound anti-CD3/anti-CD28 antibodies (2 μg/ml) or different doses of murine recombinant IL-27. For tumor infiltrating lymphocyte isolation, tumors were isolated, smashed into small pieces, and incubated at 37 °C for 1 h in the presence of Collagenase IV (Sigma-Aldrich) and DNase (Sigma-Aldrich). The tumor infiltrating lymphocytes were purified by Filcoll from single tumor cell suspension.

**Intracellular staining**. Spleen cells, CD8[+] T-cells and tumor infiltrating lymphocytes were stimulated by PMA (50 ng/ml) (Sigma-Aldrich) and Ionomycin (1 μg/ml, Sigma-Aldrich) in the presence of Golgistop (BD Bioscience) for 4 h. Cells were stained by monoclonal antibodies against CD3 (clone: 17A2, APC-780, 5 μg/ml, eBioscience), CD8 (clone: 53-6.7, PE, 5 μg/ml, BD) and CD45 (clone: 30-F11, PE-cy7, 5 μg/ml, eBioscience), fixed and permeated (BD, Cytofix/Cytoperm), followed by intracellular staining.

**Plasmids**. Mouse IL-27 p28 3′UTR plasmid was cloned by inserting p28 3′UTR into the pGL3 control vector (Promega) between XbaI and FseI sites. Primers used for amplification of p28 3′UTR were TTCTAGACACCTAGCTTCAAGCCCTA TGG (sense); and GGC CGGCCCGGGCTGGATGGCTTTATTA (anti-sense); p28 3′UTR mutants were generated with Mutagenesis kit. All plasmid DNA were prepared with QIAGEN Endo-free Maxi-Prep kits (QIAGEN).

**RNA purification and real-time RT-PCR**. Quantitative real time PCR (qRT-PCR) was performed by a modified protocol. Briefly, cDNA samples converted from 1 μg of total RNA were diluted and studied at several concentrations. Diluted cDNA was mixed with a pair of primers (10 μM)[23]. The sequences of primers were: IL-27 p28: CTCTGCTTCCTCGCTACCAC (sense), GGGGCAGCTTCTTTTCTTCT (anti-sense); Luciferase: ATTTATCGGAGTTGCAGTTGCGCC (sense), ACAAA-CACTACGGTAGGCTGCGAA (anti-sense); TNF: AGCCGATGGGTTG-TACCTTGTCTA (sense); GAGATAGCAAATCGGCTGACGGT (anti-sense).

**RNA IP**. Proteins extracted from J774 cells stimulated by LPS for 4 h were incubated with beads pre-coated with anti-TTP antibody (Catalog#: T5327, Sigma) and control IgG. After washing three times, RNA was extracted from Beads, and reverse-transcribed into cDNA, followed by detecting p28 and TNF mRNA by real-time PCR.

**ELISA**. Supernatants of cell culture, serum and ascites were stored in −70 °C freezer. IL-27, IFN-γ and TNF were detected by mouse IL-27 ELISA kit (Catalog#: 88-7274, eBioscience), mouse IFN-γ ELISA kit (Catalog#: 555138, BD Biosciences) and mouse TNF ELISA kit (Catalog#: 555268, BD Biosciences) according to the manufacturer's instructions. Concentrations were calculated by regression analysis of a standard curve.

**Transient transfection and luciferase assay**. Transient transfections were performed by electroporation. J774A.1 cells and HEK293 cells were transfected with luciferase vectors and TTP expression plasmids. Transfected cells were collected at 24 h for RNA extraction and at 48 h for measurement of luciferase activity.

**Histological analysis and Immunofluorescence staining**. Tumors were isolated and fixed in 10% formaldehyde solution. HE staining was performed on tumor sections. Tumor tissues were embedded in OCT, cut into 5 μm sections, and fixed. Non-specific binding was blocked by 5% bovine serum albumin (BSA) for 40 min. Then, sections were incubated with anti-CD8 antibody (eBioscience) overnight at 4 °C in a humidified chamber. Next, slides were incubated with AF488-labeled anti-rat IgG (1:200) for 1 h at room temperature. Nuclear counterstaining was performed with diaminidophenylindol (DAPI, 1:200 in PBS, Invitrogen). Rat IgG was used as negative control. The images were taken by fluorescence microscopy.

**Statistics**. For mouse studies, we chose a sample size of four mice per group as it provided 80% power to confirm mean differences of 2 s.d. or larger using the $p < 0.05$ significance criterion. GraphPad Prism was used for data analysis. Unpaired two-tailed Student's $t$ test was performed for comparison between two groups. One-way ANOVA (Nonparametric) with Tukey (Compare all pairs of columns) or Kruskal–Walls test with Dunn's Multiple Comparison was used for comparison among multiple groups. Data shown are mean+s.d. of at least three independent experiments. $*p < 0.05$; $**p < 0.01$; $***p < 0.001$ between indicated groups.

**Data availability**. The authors declare that data supporting the findings of this study are available within the article and its supplementary information files or are available from the corresponding author on request.

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

## Acknowledgements

We thank Perry Blackshear (National Institute of Environmental Health Sciences, Durham, NC) for providing us the TTP knockout mice and the expression vectors for CMV hTTP.Flag, CMVhTTP/C124R, CMVhTTP/C147R mutant, and thank Mike Marcinkowski (Saint Louis University) for editing. This work was supported by the National Cancer Institute of the National Institutes of Health under Award Number R01CA163808 (J.L.). The content is solely the responsibility of the authors and does not necessarily represent the official views of the National Institutes of Health.

## Author contributions

J.L. and Q.W. designed the experiments, analyzed the data and wrote the manuscript. Q.W., H.N. and H.P. performed the experiments and acquired the data. L.W., R.H. and D.F.H. revised and wrote manuscript.

## Additional information

**Competing interests:** The authors declare no competing financial interests.

