## [Peer Review File · Nature Communications]

Reviewers' Comments:

Reviewer #1 (Remarks to the Author)

Wang et al demonstrated that TTP promotes the degradation of p28 mRNA. TTP regulates and decreases IL-27 production in mononuclear phagocytes. Mononuclear phagocytes secrete IL-27 to promote a robust CTL response. At the end of this study, the authors examined the effects of TTP deficient mice in tumor development and nicely demonstrate that in the absence of TTP and IL-27R, tumor development was similar to WT mice and IL-27R deficient mice but not TTP KO mice, which have enhanced IL-27 production and CTL responses.

The reviewer thoroughly enjoyed reading this manuscript, which was well written. Experiments were relevant and sequentially logical.

Minor comment:

The authors need to state what are Zfp36^{-/-} mice—TTP KO's.

Reviewer #2 (Remarks to the Author)

There are several studies that link IL-27 to promoting CD8 T cell responses most prominently in the setting of cancer models. Here, the overall take home message is that TTP is a negative regulator of IL-27p28 and as a consequence the TTP KO mice have basal elevated CD8 responses and better anti-tumor responses is unexpected. The authors are also right when they highlight that not much thought has been given to what turns off IL-27.

Major Points

1. The authors refer to "naïve" T cells throughout the manuscript but they show in Fig 1 that they have quite a large population of CD69⁺ CD8 T cells that are already present in the KO. Since it is not clear from the description of the experiments if they truly obtained naïve (not antigen experienced cells) or used bulk CD8 populations they need to either redo many of the experiments using truly naïve cells OR they need to modify how they describe and interpret the data presented. For example, is the increased CFSE dilution they see in Fig 1 just because they start with a larger population of antigen experienced cells? This makes it difficult to interpret the data as presented.
2. At many points the authors show representative flow plots – but this is not accompanied by any further analysis that shows statistics or collated data sets. This needs to be addressed in Figs 1, 2 and 5.
3. Fig 2c/d the IFN-g production seems very, very low for a CD8 T cell population. Is this still true if they use more polarizing conditions that might include IL-12?
4. Figure 5A needs the WT control supes combined with the WSX-1KO CD8. Fig 5B seems to have a major error in labeling. Its not clear what they are gating on. The authors do note that these supes still enhance IFN-g production even in the absence of WSX-1. Since they do have an idea of what other pro-inflammatory cytokines are over produced in the cultures – can they show the impact of neutralization of TNF or IL-23 for example that have also been shown to promote IFN-g production? In other words IL-27 alone is not sufficient for this phenotype but works with other cytokines.
5. Fig 6. Is the increased resistance to the tumor cell challenge of the Zfp36 KO dependent on CD8 and/or IFN-g? This is a key point if the authors want to close the loop showing that the 27-dependent effect is due to CD8. This can be tested using depleting Abs.

6. The authors are correct that little is known about what limits IL-27 production. This itself raises the question of what regulates TTP and one obvious candidate might be IL-10. So, does IL-10 limit IL-27 or upregulate TTP? At this point, if IL-10 does not impact this pathway that is fine – but if it does, it really closes the loop on this study.

Minor Points

The Hunter JI paper on the IL-27 transgenics shows a massive activation of CD8 T cells while the Kedl PNAS paper is a good non-cancer model that illustrates the impact of endogenous IL-27 on CD8 T cell responses. Citing these manuscripts would further strengthen/broaden the overall message of the study.

The authors need to provide additional information on the ELISA kits being used. Do they detect IL-27p28 or the heterodimer. If its only p28 then the graphs need to reflect this.

Introduction – the authors might consider modifying the sentence that notes that the mechanisms that regulate the development and function of CTLs remain elusive. I think the literature is quite rich in examples of mechanisms that regulate the development and function of CTLs

Reviewer #3 (Remarks to the Author)

Very little is known about the role of zfp36 in lymphocytes. Unfortunately the work presented here does little to enlighten us. The links to IL27 are novel but unsurprising. In general, the presentation of the data is extremely poor with little or no consideration given to biological variation or proper quantitation. Some statistical tests are used but it is not clear which test is being applied to what data.

The data in Suppl Fig 1a showing CD4 and CD8 staining of the thymus is of very poor quality. Better quality data must be shown. The age and sex of the mice is an important consideration when interpreting this data and this information should be clearly provided in the figure legend. In general showing percentages is unhelpful unless we are also give the total numbers of cells as well.

Figure 1 A would be better with dot plots. To conclude anything about surface antigen expression we need to see more than the Flow cytometry. Overlay histograms can help visualize the point better. The median fluorescence intensity must be quantitated from a number of different mice and then compared. Here information on the numbers of cells in each population is essential to properly describe the phenotype.

The data leading to claims about differences in the production of cytokines etc need to be similarly analysed. Moreover if the ratio of naive to effector cells is change then the results are due to a population effect and say nothing interesting about the role of zfp36 in regulating T cell cytokines.

The isolation and activation of T cells for three days could show and effect on proliferation. To be sure of this we would need to know that the initial starting populations put into culture were equivalent. As the authors show that the population of effector cells is different in proportion this seems to be the likely explanation. No information is given on biological variation within experimental groups in this assay.

The same problems of initial different population effect and lack of biological variability exists with Figure 1D.

Figure 2 reinforces some of these criticisms with the authors own data. The lack of variability persists. Moreover the conclusion that there is no effect on cytokine mRNA or protein is premature as it is based on only a single stimulation and timepoint.

If the data shown is from a representative experiment what is n ? For the statistics why not pool

the data from the three experiments with similar results?

Fig 3 A-C is uninterpretable as we are not told what N is.

The results in Figure 4 are convincing but they are unsurprising. Any ARE containing transcript will give similar results in such as system.

Figure 5 also needs to show more than representative flow cytometry. The effects claimed need to be properly quantitated. Roles for IL27 in T cells reported by others should be discussed in this context (<https://www.ncbi.nlm.nih.gov/pubmed/25267651>).

Data in figure 6 is difficult to interpret. For example what is being shown in 6C is far from clear- we need to be clear on how many mice how many tumors are. 6D need quantitation of the effect. 6 E-F It would be better to the makeup of cellular infiltrate in absolute numbers. What are the statistics here?

Reviewer #1

The reviewer thoroughly enjoyed reading this manuscript, which was well written. Experiments were relevant and sequentially logical. Minor comment: The authors need to state what are Zfp36^{-/-} mice—TTP KO's

Response: We sincerely appreciate the thoroughly positive comments of the reviewer. The statement “TTP is an RNA binding protein encoded by *Zfp36* gene”, and “*Zfp36^{-/-}* mice are also called TTP knockout (KO) mice” are added in the text.

Reviewer #2

Major Points

1. The authors refer to “naïve” T cells throughout the manuscript but they show in Fig 1 that they have quite a large population of CD69⁺ CD8 T cells that are already present in the KO. Since it is not clear from the description of the experiments if they truly obtained naïve (not antigen experienced cells) or used bulk CD8 populations they need to either redo many of the experiments using truly naïve cells OR they need to modify how they describe and interpret the data presented. For example, is the increased CFSE dilution they see in Fig 1 just because they start with a larger population of antigen experienced cells? This makes it difficult to interpret the data as presented.

Response: We are sorry for the confusion. In this revision, we added more detailed information in figure legend on the CD8 T cells used in the experiments. Fig. 1a shows the activation status of CD8 T cells without ex vivo stimulation by gating on CD3 and CD8 double positive cells of splenocytes from WT and *Zfp36^{-/-}* mice. In the previous Fig. 1C (now shown as Fig. 1d in the revision), naïve CD8⁺ T cells were used to show the effects of TTP on CD8 T cell proliferation. The untouched naïve CD8 T cells were isolated by EasySep™ Mouse Naïve CD8⁺ T Cell Negative Isolation Kit purchased from Stemcell Technologies (Catalog#: 19858) and the purity of naïve CD8 T cells is up to 98%.

2. At many points the authors show representative flow plots – but this is not accompanied by any further analysis that shows statistics or collated data sets. This needs to be addressed in Figs 1, 2 and 5.

Response: A similar concern was also raised by the third reviewer. We have performed statistical analyses to all flow plots. The results of statistical analyses are now added in Figures 1, 2 and 5, as well as in Supplementary Figures 1, 2 and 4.

3. Fig 2c/d the IFN- γ production seems very, very low for a CD8 T cell population. Is this still true if they use more polarizing conditions that might include IL-12?

Response: Yes, we performed new experiments to stimulate naïve CD8 T cells with IL-12 in the presence of anti-CD3/CD28 Abs. As shown in new Supplementary Fig. 2b, IL-12 significantly increased IFN- γ production at similar levels in WT and *Zfp36^{-/-}* CD8 T cells. Furthermore, the production of IFN- γ was similar between WT and *Zfp36^{-/-}* CD8 T cells in response to PMA and Ionomycin stimulation as well as to TCR activation (new Supplementary Fig. 2c). These data further support the conclusion that CD8 T cells deficient of TTP have no intrinsic defect in IFN- γ production.

4. Figure 5A needs the WT control supes combined with the WSX-1KO CD8. Fig 5B seems to have a major error in labeling. Its not clear what they are gating on. The authors do note that these supes still enhance IFN- γ production even in the absence of WSX-1. Since they do have an idea of what other pro-inflammatory cytokines are over produced in the cultures – can they show the impact of neutralization of TNF or IL-23 for example that have also been shown to promote IFN- γ production? In other words IL-27 alone is not sufficient for this phenotype but works with other cytokines.

Response: We performed new experiments by including the WT control supernatants combined with the WSX-1^{-/-} CD8 T cells and obtained similar results. The increased IFN- γ production by TTP^{-/-} supernatants was significantly reduced when culturing with WSX-1^{-/-} CD8 T cells, while little difference in IFN- γ production between WT and WSX-1^{-/-} CD8 T cells when culturing with WT control supernatants (new Fig.

5a). Since there are still enhanced IFN- γ production by TTP^{-/-} supernatants even in the absence of WSX-1, as suggested by the reviewer, other pro-inflammatory cytokines, such as TNF- α and IL-23, may promote IFN- γ production in addition of IL-27. To answer this question, we performed new experiments by treating the TTP^{-/-} supernatants with TNF- α and IL-23 neutralizing antibody prior to culturing it with WT naïve CD8 T cells. Neutralizing TNF- α and IL-23 reduced the IFN- γ production in WT CD8 T cells induced by the TTP^{-/-} supernatants, with a little stronger effects of TNF- α blockade than IL-23 blockade (new Fig. 5b). Though neutralizing TNF- α and IL-23 reduced IFN- γ production, there still remained a large amount of IFN- γ production compared with it in WSX-1^{-/-} CD8 T cells culturing with the TTP^{-/-} supernatants, suggesting that IL-27 plays a major role, while TNF- α and IL-23 play a minor role, in inducing IFN- γ production in CD8 T cells mediated by TTP^{-/-} supernatants. It also suggests that IL-27 can induce IFN- γ production in CD8 T cells; other cytokines (such as TNF- α and IL-23) may work with IL-27 to further enhance IFN- γ production.

The data shown in the previous Fig. 5B was gated on CD4 negative T cells. In this revision, we replaced it with new data by gating directly on CD8⁺ T cells (now shown as Fig. 5c).

5. Fig 6. Is the increased resistance to the tumor cell challenge of the *Zfp36* KO dependent on CD8 and/or IFN-g? This is a key point if the authors want to close the loop showing that the 27-dependent effect is due to CD8. This can be tested using depleting Abs.

Response: We thank the reviewer for the suggestion on using depletion Ab. We performed new animal experiments using the antibody for CD8 T cell depletion (*InVivo*MAB anti-mouse CD8 α Clone: 53-6.72 purchased from Bio X cell). The CD8 T cell depleting antibody was administrated five times in total (on day 0, 4, 8, 12 and 15) after tumor cell inoculation. In consistent with the results in Fig. 6a, *Zfp36*^{-/-} mice showed retarded tumor growth compared with WT control mice (new Fig. 6j,k). Importantly, depletion of CD8 T cells in *Zfp36*^{-/-} mice resulted in acceleration of tumor growth, even a little faster than it in WT mice with CD8 depletion (new Fig. 6j,k). The efficiency of CD8 T cell depletion was checked on day 18 after tumor cell inoculation. The percentages of CD8 T cells were significantly reduced by the CD8 depleting antibody (new Supplementary Fig. 4b). These data close the loop showing that IL-27-dependent antitumor effect is mediated by CD8 T cells in *Zfp36*^{-/-} mice.

6. The authors are correct that little is known about what limits IL-27 production. This itself raises the question of what regulates TTP and one obvious candidate might be IL-10. So, does IL-10 limit IL-27 or upregulate TTP? At this point, if IL-10 does not impact this pathway that is fine – but if it does, it really closes the loop on this study.

Response: We performed new experiments by stimulating WT macrophages with LPS, IL-10 or both, then measured p28 and EBI3 mRNA expression. As reported previously, IL-10 inhibited the p28 (new Supplementary Fig. 5a) but not EBI3 (new Supplementary Fig. 5b) mRNA expression induced by LPS. In the same setting, LPS strongly induced TTP expression which is in line with our previous report (new Supplementary Fig. 5c). While IL-10 induced minimal level of TTP expression by itself, it inhibited LPS-induced TTP expression (new Supplementary Fig. 5c). IL-10 does not upregulate TTP in activated macrophages, suggests that IL-10-mediated suppression of IL-27 is independent of TTP. Moreover, we treated WT and *Zfp36*^{-/-} BMDMs with LPS in the presence or absence of IL-10 and then measured p28 mRNA expression. As shown in Supplementary Fig. 5d, IL-10 inhibited p28 expression in both BMDMs, indicating that IL-10 does not impact this pathway.

Minor Points

7. The Hunter JI paper on the IL-27 transgenics shows a massive activation of CD8 T cells while the Kedl PNAS paper is a good non-cancer model that illustrates the impact of endogenous IL-27 on CD8 T cell responses. Citing these manuscripts would further strengthen/broaden the overall message of the study.

Response: We are sorry for missing the two important papers related to the impact of IL-27 on CD8 T cells. The two papers are cited in the revised manuscript.

8. *The authors need to provide additional information on the ELISA kits being used. Do they detect IL-27p28 or the heterodimer. If its only p28 then the graphs need to reflect this.*

Response: The ELISA kit used in this study is to measure the heterodimer. We added more detailed information about the ELISA kits in this revision (mouse IL-27 ELISA kit. Catalog#: 88-7274, eBioscience).

9. *Introduction – the authors might consider modifying the sentence that notes that the mechanisms that regulate the development and function of CTLs remain elusive. I think the literature is quite rich in examples of mechanisms that regulate the development and function of CTLs.*

Response: We sincerely appreciate the advice and have modified the sentence in the text.

Reviewer #3

1. *Very little is known about the role of zfp36 in lymphocytes. Unfortunately the work presented here does little to enlighten us. The links to IL27 are novel but unsurprising. In general, the presentation of the data is extremely poor with little or no consideration given to biological variation or proper quantitation. Some statistical tests are used but it is not clear which test is being applied to what data.*

Response: We are sorry for the confusion and sincerely appreciate the comments on biological variation. We have performed statistical analyses on all data in this revised manuscript. The methods of statistical analysis, variation and sample size are now included in figure legends.

2. *The data in Suppl Fig 1a showing CD4 and CD8 staining of the thymus is of very poor quality. Better quality data must be shown. The age and sex of the mice is an important consideration when interpreting this data and this information should be clearly provided in the figure legend. In general showing percentages is unhelpful unless we are also give the total numbers of cells as well.*

Response: We performed new staining of the CD4 and CD8 T cells in thymus and obtained similar results. Both the percentages and total numbers of CD4 and CD8 T cells in thymus are comparable between WT and *Zfp36*^{-/-} mice (new Supplementary Fig. 1a,b). The WT and *Zfp36*^{-/-} mice used in the study are female aged 6-8 weeks old. This information is added in the figure legend.

3. *Figure 1 A would be better with dot plots. To conclude anything about surface antigen expression we need to see more than the Flow cytometry. Overlay histograms can help visualize the point better. The median fluorescence intensity must be quantitated from a number of different mice and then compared. Here information on the numbers of cells in each population is essential to properly describe the phenotype.*

Response: We thank the reviewer for the invaluable advice. In this revision, we provide the flow data with dot plots (new Fig. 1a) and with histogram (new Supplementary Fig. 1d). The median fluorescence intensities from three different mice are quantitated and shown in new Supplementary Fig. 1d. The numbers of CD8 T cells in each population are also quantitated and shown as new Supplementary Fig. 1c. All data support the conclusion that there are more effector CD8 T cells in *Zfp36*^{-/-} mice compared to WT mice.

4. *The data leading to claims about differences in the production of cytokines etc need to be similarly analysed. Moreover if the ratio of naïve to effector cells is change then the results are due to a population effect and say nothing interesting about the role of zfp36 in regulating T cell cytokines.*

Response: We have re-analyzed the difference in cytokine production and presented the quantitative results in new Fig. 1c. *Zfp36*^{-/-} CD8 T cells produced more IFN- γ and TNF- α , but similar levels of IL-2, granzyme B and perforin compared with WT CD8 T cells (Fig. 1b,c). As the reviewer pointed out, the effector CD8 T cells are increased in *Zfp36*^{-/-} mice (Fig. 1a-c and Supplementary Fig. 1c,d). To exclude the inference of effector cells, we used the purified naïve CD8 T cells in the subsequent experiments, and found that there was no intrinsic defect in producing IFN- γ by *Zfp36*^{-/-} CD8 T cells ex vivo (Fig. 2a-d and Supplementary Fig. 2a-d). The increased CTL in *Zfp36*^{-/-} mice is mediated by IL-27 as confirmed with the double knockout mice (Fig. 5a-e).

5. *The isolation and activation of T cells for three days could show an effect on proliferation. To be sure of this we would need to know that the initial starting populations put into culture were equivalent. As the authors show that the population of effector cells is different in proportion this seems to be the likely explanation. No information is given on biological variation within experimental groups in this assay.*

Response: We are sorry for the confusion. Since there are more effector CD8 T cells in *Zfp36*^{-/-} mice, for fair comparison we isolated naïve CD8 T cells from spleens of WT and *Zfp36*^{-/-} mice, used same number of naïve CD8 T cells for ex vivo stimulation with plate-coated anti-CD3/CD28 Abs, and then performed CFSE dilution assay and Ki67 detection on day three. The levels of CFSE and Ki67 at the beginning of culture (0h) are now included in new Fig. 1d. The results of statistical analysis are also included in new Fig. 1d as well.

6. *The same problems of initial different population effect and lack of biological variability exists with Figure 1D.*

Response: Same number of naïve CD8 T cells isolated from WT and *Zfp36*^{-/-} mice was used in the previous Figure 1D. The levels of PI⁺ and Annexin-v⁺ CD8 T cells at the beginning of culture (0h) and the results of statistical analysis are shown in new Fig. 1e. The representative dot plots are shown as new Supplementary Fig. 1e.

7. *Figure 2 reinforces some of these criticisms with the authors own data. The lack of variability persists. Moreover the conclusion that there is no effect on cytokine mRNA or protein is premature as it is based on only a single stimulation and time point.*

Response: The initial status of the naïve CD8 T cells (0h) is now included in Fig. 2a. The results of statistical analyses are shown in new Fig. 2b,d. To further confirm the effects of TTP on cytokine production, we performed new experiments by stimulating the naïve CD8 T cells with PMA/Ionomycin and IL-12, in addition of TCR activation. The production of IFN- γ was similar between WT and *Zfp36*^{-/-} CD8 T cells in response to different stimuli at different time points (new Supplementary Fig. 2b,c). These data support the conclusion that CD8 T cells deficient of TTP have no intrinsic defect in IFN- γ production.

8. *If the data shown is from a representative experiment what is n ? For the statistics why not pool the data from the three experiments with similar results?*

Response: We pooled the data from three experiments (n=3) and performed statistical analysis. The quantitative results are now included in new Fig. 2b,d, and in Supplementary Fig. 2d.

9. *Fig 3 A-C is uninterpretable as we are not told what N is.*

Response: A detailed description including the N and statistics are now included in figure legend.

10. *The results in Figure 4 are convincing but they are unsurprising. Any ARE containing transcript will give similar results in such as system.*

Response: Though TTP is known to bind to ARE in the 3'UTR, our data demonstrate that not all AREs are used by TTP to exert its function. TTP promotes p28 mRNA decay through several AREs but not ARE2. This inhibitory effect is mediated by the two zinc fingers in TTP protein. To the best of our knowledge, this is the first work showing how IL-27 is down-regulated by RNA-binding protein TTP.

11. *Figure 5 also needs to show more than representative flow cytometry. The effects claimed need to be properly quantitated. Roles for IL27 in T cells reported by others should be discussed in this context (<https://www.ncbi.nlm.nih.gov/pubmed/25267651>).*

Response: Quantitative results of the flow cytometry data are now included in Fig. 5. We are sorry for missing the important paper and have cited it in the revised manuscript.

12. *Data in figure 6 is difficult to interpret. For example what is being shown in 6C is far from clear-we need to be clear on how many mice how many tumors are. 6D need quantitation of the effect. 6 E-F It would be better to the makeup of cellular infiltrate in absolute numbers. What are the statistics here?*

Response: The data shown in Fig. 6c is the quantitative results of the images shown in Fig. 6b, which includes four mice and four tumors in each group. To better visualize the infiltrates, we stained CD8 T cells on different tumor sections and quantitated the number of tumor-infiltrating CD8 T cells in each group. More infiltrating CD8 T cells were observed per field in tumors of *Zfp36*^{-/-} mice compared with tumors of other groups; while the increased CD8 T cells were reduced significantly in tumors of the DKO mice (new Fig. 6d,e). Because the tumor mass (sizes) are markedly different among four groups (Fig. 6a-c), for fair comparison we normalized the infiltrates against tumor mass. The percentages and number of CD8 T cells in tumors are quantitated and shown in Fig. 6f-i. One-way ANOVA (and Nonparametric) with Tukey was used to analyze the difference among four groups. ***: $p < 0.001$; **: $p < 0.01$; *: $p < 0.05$ between indicated groups. In addition, we performed new *in vivo* experiments using CD8 depletion antibody. In consistent with the results shown in Fig. 6a, *Zfp36*^{-/-} mice showed retarded tumor growth compared with WT control mice (new Fig. 6j,k). Importantly, depletion of CD8 T cells in *Zfp36*^{-/-} mice resulted in acceleration of tumor growth (new Fig. 6j,k). Taken together, these data confirm that IL-27-dependent antitumor effect is mediated by CD8 T cells in *Zfp36*^{-/-} mice.

Reviewers' Comments:

Reviewer #2 (Remarks to the Author)

The authors have made several revisions to the manuscript that help close the loop (i.e. the anti-CD8 experiment on Fig 6) and have added in many additional experimental details that help clarify how the experiments are performed. These clarifications themselves are useful but have raised a couple of questions that need to be considered - specifically about cell intrinsic vs extrinsic effects of TTP that just need to be thought through. In addition, there remain several gaps between the last reviews and the revision that also need to be addressed.

1. Figure 1D shows that purified naive CD8 T cells from Zfp36KO mice have an increased ability to proliferate. This implies that there is a T cell intrinsic role for this transcription factor on the regulation of T cell proliferation and survival. This stands in contrast to the author's conclusion that TTP modulates CD8 T cell activation in a T- cell extrinsic fashion. This looks like a major effect and (as noted by reviewer 2) there are concerns about the data previously presented and the authors need to provide data on the numbers of T cells not just percentages in the later figures. This was requested but has not been addressed adequately.

2. Figure 2A - needs cell numbers not just %. Same issue in Fig 5 panels A, B and D.

3. Figure 5. The authors show that supernatants from KO macrophages promote T cell production of IFN-g. There was a previous suggestion that they neutralize over cytokines that might affect this process and they have neutralized TNF and IL-23 and see a modest effect. The details of the Abs are not apparent in the materials and methods and it seems like it is critical to know if the authors are blocking IL-23 p19 or the p40 subunit shared with IL-12. IL-12 seems like the most likely candidate for a macrophage derived cytokine that promotes IFN-g and it may or may not act in concert with IL-27.

Reviewer #3 (Remarks to the Author)

The revised manuscript attempts to clarify what role TTP has in the cellular immune response to tumors and concludes a major role for the IL27 axis but no intrinsic role in CD8T cell activation. The enhanced anti-tumor immunity of TTP KO mice is not evident when IL27 signaling is abolished in WSX-1 mutant mice, or upon depletion of CTL with anti-CD8 antibodies. I think the conclusion that IL27 overproduction by TTP KO macrophages affects CD8 T cells is likely to be correct and that loss of IL-27 signaling or depletion of CD8 T cells will diminish anti-tumor immunity. I would have been surprised if either of these manipulations did not diminish anti-tumor immunity. I think the conclusion that TTP has no role in CD8 T cells is likely to be incorrect and I ask the authors to carefully reconsider this conclusion.

The conclusion stated on line 132 that "TTP modulates CD8 T cell activation in a T cell extrinsic fashion." is not supported by the authors data. The authors own results (data in Fig1 d and e) contradict this conclusion. It would appear the authors base their conclusion on the observation that IFN γ is not different at the time tested, but to extrapolate from this specific instance to such a strong general conclusion is unwarranted.

New thymic CD4/8 staining data supplied in supplemental Fig1a is a result of improved quality, but what is stated in the figure legend is what is shown in the figure are not compatible. If the gating is on CD3+ cells how do the authors explain the presence of the CD4-CD8- cells? Something isn't right here but it is not clear what the problem might be. This could be fixed by showing the gating

strategies and populations under scrutiny.

From what is shown in Figure 1a it is uninformative to state that CD8 cells express higher levels of activation markers because the populations being compared are not the same. Separation of naive and memory subsets is required then the median fluorescence intensity should be compared. I do not think that this is what has been done in supplementary Fig1d. Showing cell % is uninformative and misleading and could well be removed. Showing activation marker expression on naive and memory cells could be helpful if there is a difference and the biological significance of the difference were stated. In Fig1 b the differential cytokine production from a mixed population is not very informative reflecting again the comparison being between mixtures of cells that differ in the proportions of naive and activated memory cells.

The proliferation data is of interest but the analysis is rather superficial. It is possible to determine cell number at each division and the display of % CFSE low tells us nothing quantitatively about the number of divisions that appears from the figure to be different. This is potentially interesting but is not developed well enough for publication. It isn't clear whether this has any in vivo relevance to the tumor immunity results.

What is the justification to use quadrants to gate in figure 2a and how are positive and negative thresholds for setting gates in the flow cytometry analysis determined? In 2b some attempt at quantitation is made but it looks like there is no effect. The text accompanying this figure suggests there may be a decrease. This could be more precisely described.

The new data on IFN γ mRNA are interesting but this study is still incomplete. Why not measure other cytokines such as TNF? Also the accumulation of cytokines by ELISA over a timecourse? In supplemental figure 2d the nature of the cell mixture is unclear and the means of stimulation so strong, and limited to a single timepoint, that it is hard to be convinced that IFN γ production by CD8+ T cells under physiological conditions of T cell stimulation is TTP independent.

The addition of supernatants from activated macrophages to the CD8T cells and the use of receptors deficient cells does indicate IL27 plays a role but the point is somewhat labored and other factors are also active in those sups.

From 6a I am not convinced that it is correct to say tumors grow more quickly in WSX or DKO mice. For the first 20 days no clear difference is evident. Are there other data that back this conclusion up?

Is the % of DKO CD8 in figure 6h less than in the zfp36 single KO?

The identification of IL-27 as a TTP target is not novel and has been found in the major analysis of TTP targets in macrophages from Kovarik's group. This can be found on line in their web tool that navigates through the data. This needs to be acknowledged.

<http://ttp-atlas.univie.ac.at/>

There are many spelling and grammatical errors throughout the manuscript that will need to be corrected.

1. “...whether the effects of TTP are intrinsic or extrinsic to CD8 T cells...”

Response: We greatly appreciate both reviewers’ thoughtful concern on “whether the effects of TTP are intrinsic or extrinsic to CD8 T cells”. This statement was made based on the comparable production of IFN- γ between activated WT and Zfp36^{-/-} CD8 T cells. Indeed, the proliferation and survival of naïve Zfp36^{-/-} CD8 T cells were increased compared to naïve WT CD8 T cells (Fig. 1d,e & supplementary Fig. 1f,g), indicating a T cell intrinsic effect of TTP on cell proliferation and survival. When naïve Zfp36^{-/-} CD8 T cells were activated and become effector cells, they expressed similar levels of activation markers and produced comparable amounts of TNF- α and IFN- γ as the activated WT CD8 T cells (Fig. 2 & supplementary Fig. 2), suggesting a T cell extrinsic role for TTP on cytokine production in CD8 T cells. Therefore, to more accurately reflect the results we have revised the statement as “TTP regulates CD8 T cell proliferation and survival in an intrinsic manner, while modulates cytokine production in an extrinsic fashion.”

Reviewer #2

1. *Figure 1D shows that purified naive CD8 T cells from Zfp36KO mice have an increased ability to proliferate. This implies that there is a T cell intrinsic role for this transcription factor on the regulation of T cell proliferation and survival. This stands in contrast to the author's conclusion that TTP modulates CD8 T cell activation in a T-cell extrinsic fashion. ...the authors need to provide data on the numbers of T cells not just percentages in the later figures. This was requested but has not been addressed adequately.*

Response: Please refer to the above response to “A concern raised by both reviewers”. We now include the numbers of T cells in Figure 1D and in other figures wherever applicable.

2. *Figure 2A - needs cell numbers not just %. Same issue in Fig 5 panels A, B and D.*

Response: The summaries of cell numbers in Figure 2A and in Figures 5A, B and D are now included as Figure 2b and in Figure 5a, c & d, respectively. The percentages of cells shown in the previous Figure 5a were moved to supplementary Figure 4a due to a limitation of space.

3. Figure 5. The authors show that supernatants from KO macrophages promote T cell production of IFN- γ . There was a previous suggestion that they neutralize over cytokines that might affect this process and they have neutralized TNF and IL-23 and see a modest effect. The details of the Abs are not apparent in the materials and methods and it seems like it is critical to know if the authors are blocking IL-23 p19 or the p40 subunit shared with IL-12. IL-12 seems like the most likely candidate for a macrophage derived cytokine that promotes IFN- γ and it may or may not act in concert with IL-27.

Response: The neutralizing antibody used was directed specifically against the unique subunit of IL-23 p19. Detailed information about the antibodies are now included in the Cells and reagents section of Methods.

As mentioned by the reviewer, IL-12 is known to be a strong activator for CD8 T cells. To determine whether other cytokines (such as IL-12, TNF- α and IL-23) may act with or without IL-27 to activate CD8 T cells, we performed new experiments by co-culturing the WSX-1^{-/-} CD8 T cells with supernatants from activated Zfp36^{-/-} macrophages in the presence of neutralizing antibodies against IL-12/IL-23 p40, TNF- α , and IL-23 p19. In the absence of IL-27 signaling, neutralizing IL-12 and TNF- α , but not IL-23, inhibited the production of IFN- γ in WSX-1^{-/-} CD8 T cells (Fig. 5b,c), indicating that IL-12 and TNF- α induce IFN- γ production in CD8 T cells independent of IL-27. Taken together, our data demonstrate that IL-27 promotes IFN- γ production in CD8 T cells mediated by TTP^{-/-} supernatants. Other cytokines, such as IL-12 and TNF- α , may work with IL-27 to further enhance IFN- γ production.

Reviewer #3

1. I think the conclusion that TTP has no role in CD8 T cells is likely to be incorrect and I ask the authors to carefully reconsider this conclusion. The conclusion stated on line 132 that “TTP modulates CD8 T cell activation in a T cell extrinsic fashion.” is not supported by the authors data. The authors own results (data in Fig1 d and e) contradict this conclusion. It would appear the authors base their conclusion on the observation that IFN γ is not different at the time tested, but to extrapolate from this specific instance to such a strong general conclusion is unwarranted.

Response: We sincerely appreciate the insightful comment by the reviewer. Please refer to the above response to “A concern raised by both reviewers.”

2. New thymic CD4/8 staining data supplied in supplemental Fig1a is a result of improved quality, but what is stated in the figure legend is what is shown in the figure are not compatible. If the gating is on CD3+ cells how do the authors explain the presence of the CD4-CD8- cells? Something isn't right here but it is not clear what the problem might be. This could be fixed by showing the gating strategies and populations under scrutiny.

Response: We apologize for the mislabeling. Data shown in supplemental Fig. 1a was gated on thymic cells and then checked for CD4 and CD8 expression. We have corrected the mislabeling.

3. From what is shown in Figure 1a it is uninformative to state that CD8 cells express higher levels of activation markers because the populations being compared are not the same. Separation of naïve and memory subsets is required then the median fluorescence intensity should be compared. I do not think that this is what has been done in supplementary Fig1d. Showing cell % is uninformative and misleading and could well be removed. Showing activation marker expression on naïve and memory cells could be helpful if there is a difference and the biological significance of the difference were stated.

Response: We separated the naïve (CD62L⁺CD44⁻) and memory (CD62L⁺CD44⁺) CD8 T cells and compared median fluorescence intensity (MFI) of the activation markers. The MFI levels of CD44, CD25, and CD69 were similar between naïve or memory WT and Zfp36^{-/-} CD8 T cells (supplementary Fig. 1d), which is consistent with our conclusion that the numbers of effector Zfp36^{-/-} CD8 T cells are increased. As suggested by the reviewer, we have deleted the previous supplementary Fig. 1d.

4. In Fig. 1b the differential cytokine production from a mixed population is not very informative

reflecting again the comparison being between mixtures of cells that differ in the proportions of naïve and activated memory cells.

Response: The effector CD8 T cells in the mixed population were increased in *Zfp36*^{-/-} mice (Fig. 1b,c). Since cytokines are mainly produced by effector cells, for a fair comparison, we analyzed cytokine production only in CD44⁺ activated memory CD8 T cells. The percentages of cytokine-producing cells among activated memory cells were comparable between WT and *Zfp36*^{-/-} CD8 T cells (upper panel in supplementary Fig. 1e). The number of CD44⁺ activated memory cells producing IFN- γ and TNF- α were higher in *Zfp36*^{-/-} CD8 T cells than in WT CD8 T cells (lower panel in supplementary Fig. 1e). These data confirm that the increased cytokine production in *Zfp36*^{-/-} CD8 T cells is due to an increase in number of effector cells, whereas the capacity for cytokine production by *Zfp36*^{-/-} CD8 T cells is similar to WT cells.

5. The proliferation data is of interest but the analysis is rather superficial. It is possible to determine cell number at each division and the display of % CFSE low tells us nothing quantitatively about the number of divisions that appears from the figure to be different. This is potentially interesting but is not developed well enough for publication. It isn't clear whether this has any in vivo relevance to the tumor immunity results.

Response: We performed new experiments to count the number of CFSE low cells in each division for seven generations. The difference in cell numbers reached statistical significance starting at the third generation (G3) and continued throughout the sixth generation (G6) (supplementary Fig. 1f). For relevance to tumor immunity, we reason that the increased proliferation plus the enhanced IFN- γ production in *Zfp36*^{-/-} CD8 T cells work together contributing to the TTP-mediated tumor immunity.

6. What is the justification to use quadrants to gate in figure 2a and how are positive and negative thresholds for setting gates in the flow cytometry analysis determined?

Response: We used quadrants in figure 2a to separate effector, memory and naïve cells defined by surface marker CD69, CD25, CD44, and CD62L. The negative thresholds were set with isotype controls and positive thresholds set with antibodies in activated cells (see included Fig. 1).

7. In 2b some attempt at quantitation is made but it looks like there is no effect. The text accompanying this figure suggests there may be a decrease. This could be more precisely described. The new data on IFN γ mRNA are interesting but this study is still incomplete. Why not measure other cytokines such as TNF? Also the accumulation of cytokines by ELISA over a time course?

Response: We now describe in detail the data shown in the previous Fig. 2b. When naïve CD8 T cells were activated, the percentages (Fig. 2a,c) and numbers of (Fig. 2b) of CD69⁺, CD25⁺ and CD44⁺ CD8 T cells become comparable between WT and *Zfp36*^{-/-} CD8 T cells. The MFIs of CD44 were also similar between activated WT and *Zfp36*^{-/-} CD8 T cells over a period of 48 hours (supplementary Fig. 2a). In addition, we measured TNF- α mRNA and found that, similar to IFN- γ , the expression of TNF- α mRNA did not differ between WT and *Zfp36*^{-/-} naïve CD8 T cells in response to PMA/Ionomycin and TCR stimulation at different time points (Supplementary Fig. 2c). The levels of the accumulated IFN- γ and TNF- α in supernatants were also comparable between WT and *Zfp36*^{-/-} CD8 T cells up to 48 hours (Fig. 2e).

8. In supplemental figure 2d the nature of the cell mixture is unclear and the means of stimulation so strong, and limited to a single time point, that it is hard to be convinced that IFN γ production by CD8⁺ T cells under physiological conditions of T cell stimulation is TTP independent.

Response: We performed new experiments to activate the T cells with either P/I or anti-CD3/CD28 antibodies for different periods. The percentages and cell numbers of IFN- γ -producing CD8 T cells were

similar between WT mice and the T cell-specific TTP conditional knockout mice (supplementary Fig. 2d). These additional data are in agreement with the statement that IFN- γ production in CD8 T cells is mediated by TTP extrinsically.

9. The addition of supernatants from activated macrophages to the CD8T cells and the use of receptors deficient cells does indicate IL27 plays a role but the point is somewhat labored and other factors are also active in those sups.

Response: The IL-27 receptor WSX-1^{-/-} CD8 T cells represent a unique tool that helps us uncover a new regulatory pathway for IL-27/CD8 activation mediated by TTP. With the WSX-1 deficient cells plus neutralizing antibodies against IL-12/IL-23 p40, TNF- α and IL-23 p19, we were able to define the effects of this pathway on CD8 T cell activation, the involvement of other cytokines regulated by TTP in IFN- γ production by CD8 T cells, and their consequence for tumor immunity.

10. From 6a I am not convinced that it is correct to say tumors grow more quickly in WSX or DKO mice. For the first 20 days no clear difference is evident. Are there other data that back this conclusion up?

Response: The tumor growth in WSX-1^{-/-} and DKO mice were accelerated after 20 days. We had to terminate the animal experiments at day 24 in order to comply with the guidelines of IACUC protocol. We analyzed the differences in tumor sizes at day 24 among four groups with one-way ANOVA and Tukey's multiple comparison test, and the results were statistically significant comparing all groups except between WSX-1^{-/-} and DKO mice (Fig. 6a). The weights of tumors also show a difference between WSX-1^{-/-} or DKO mice and WT mice (Fig. 6c).

11. Is the % of DKO CD8 in figure 6h less than in the zfp36 single KO?

Response: Yes, the percentages of CD8 T cells were lower in DKO mice than in single Zfp36^{-/-} mice with $p < 0.05$ (Fig. 6h).

12. The identification of IL-27 as a TTP target is not novel and has been found in the major analysis of TTP targets in macrophages from Kovarik's group. This can be found on line in their web tool that navigates through the data. This needs to be acknowledged.

Response: We navigated through the link shown in Kovarik's website, and noticed that their transcriptome-wide study identified TTP binding motifs in 498 genes. Though IL-27 is on the list, their work did not show IL-27 mRNA is targeted by TTP. To the best of our knowledge, our study for the first time reveals a direct role for TTP in regulation of IL-27 expression, in activation of CD8 T cells, and tumor immunity. Nevertheless, we appreciate the work done by Kovarik's group in providing the useful TTP binding site atlas and mention their web tool in the Discussion section.

13. There are many spelling and grammatical errors throughout the manuscript that will need to be corrected.

Response: We have requested an English editor to proof-reading the revised manuscript.

We sincerely hope that these changes are clear and satisfactory enough to permit the acceptance of this manuscript for publication by Nature Communications.

Thank you very much for your consideration.

Sincerely,

Jianguo Liu, M.D., Ph.D.

Associate Professor of Medicine

Reviewers' Comments:

Reviewer #2:

Remarks to the Author:

Concerns are addressed.

Reviewer #3:

Remarks to the Author:

The authors conclude: "suggesting that TTP regulates CD8 T cell proliferation and survival in an intrinsic manner, while modulates cytokine production in an extrinsic fashion."

The first assertion is supported by the in vitro data. But the data shows that cytokine production is not being modulated but that the number of cytokine producing CD8+ cells is increased when TTP has been knocked out. We need to be clear through whether we are dealing with increased cytokine production (on a per cells basis) or and increased number of cytokine producing cells. The written explanation of the data is still unclear and its interpretation confusing. What is the ratio of naïve to activated/effector CD8+ in the CD4cre model? Can this help clarify the point that the authors are trying to make?

Response to Reviewer #3

1. The authors conclude: “suggesting that TTP regulates CD8 T cell proliferation and survival in an intrinsic manner, while modulates cytokine production in an extrinsic fashion.”

The first assertion is supported by the in vitro data. But the data shows that cytokine production is not being modulated but that the number of cytokine producing CD8⁺ cells is increased when TTP has been knocked out. We need to be clear through whether we are dealing with increased cytokine production (on a per cells basis) or and increased number of cytokine producing cells. The written explanation of the data is still unclear and its interpretation confusing. What is the ratio of naïve to activated/effector CD8⁺ in the CD4cre model? Can this help clarify the point that the authors are trying to make?

Response: The increased cytokine production in CD8 T cells deficient in TTP is due to an increase in effector cells but not due to the difference in cytokine production. This conclusion is supported by the data shown in Figure 2 and Supplementary Figure 2. In addition, we have checked mean fluorescence intensity (MFI) of IFN- γ in activated WT and *Zfp36*^{-/-} CD8 T cells shown in the existing data in Supplementary Figure 2a, and found that the MFIs of IFN- γ were similar between activated/effector WT and *Zfp36*^{-/-} CD8 T cells (4702 \pm 224.7 vs. 4802.5 \pm 261.3; *n*=4) when gated on the same number of CD44⁺ cells. The ratio of naïve to effector CD8 T cells was similar between WT and CD4 conditional TTP KO mice (Supplementary Fig. 2e). These data demonstrate that cytokine production at per cell level is comparable between activated/effector WT and *Zfp36*^{-/-} CD8 T cells. These statements have been added to the text. We have also modified the statement in Line 148-149 as “These results demonstrate that CD8 T cell activation is not altered by TTP deficiency, and cytokine production at the per cell level is similar between activated WT and *Zfp36*^{-/-} CD8 T cells”.